# Selective activator of human ClpP triggers cell cycle arrest to inhibit lung squamous cell carcinoma

Lin-Lin Zhou [1,2,11], Tao Zhang [1,11], Yun Xue [3,4,11], Chuan Yue [5,1], Yihui Pan [2,6], Pengyu Wang [6], Teng Yang[1,6], Meixia Li [7], Hu Zhou [2,6,8], Kan Ding [2,7], Jianhua Gan [9], Hongbin Ji [3,4,10] ✉ & Cai-Guang Yang [1,2,5,6] ✉

Chemo-activation of mitochondrial ClpP exhibits promising anticancer properties. However, we are currently unaware of any studies using selective and potent ClpP activators in lung squamous cell carcinoma. In this work, we report on such an activator, ZK53, which exhibits therapeutic effects on lung squamous cell carcinoma in vivo. The crystal structure of ZK53/ClpP complex reveals a π-π stacking effect that is essential for ligand binding selectively to the mitochondrial ClpP. ZK53 features on a simple scaffold, which is distinct from the activators with rigid scaffolds, such as acyldepsipeptides and imipridones. ZK53 treatment causes a decrease of the electron transport chain in a ClpP-dependent manner, which results in declined oxidative phosphorylation and ATP production in lung tumor cells. Mechanistically, ZK53 inhibits the adenoviral early region 2 binding factor targets and activates the ataxia-telangiectasia mutated-mediated DNA damage response, eventually triggering cell cycle arrest. Lastly, ZK53 exhibits therapeutic effects on lung squamous cell carcinoma cells in xenograft and autochthonous mouse models.

Mitochondria are best known as the powerhouse of eukaryotic cells for ATP production through oxidative phosphorylation (OXPHOS)[1]. The electron transport chain (ETC), consisting of four multiprotein complexes (CI-IV), is the primary executor of OXPHOS. The ETC receives electrons from the tricarboxylic acid cycle and creates a proton gradient to drive ATP production, a biological process that relies on the maintenance of proteome homeostasis in mitochondria. The mitochondrial caseinolytic protease P (HsClpP), a serine protease in the matrix, is responsible for the homeostasis of mitochondria proteome by degrading misfolded and damaged proteins under the physiological regulation of the AAA+ ATPase chaperone ClpX[2,3]. By consuming ATP, ClpX recognizes, unfolds, and translocates the tagged proteins into the cylindrical chamber formed by the HsClpP tetradecamer for protein proteolysis[2]. Given that the inhibition of OXPHOS by the small-molecule inhibitors of ETC complexes has shown therapeutic effects in multiple cancer types[4], uncontrollable degradation of ETC subunits by

[1]State Key Laboratory of Drug Research, Centre for Chemical Biology, Shanghai Institute of Materia Medica, Chinese Academy of Sciences, Shanghai 201203, China. [2]University of Chinese Academy of Sciences, 100049 Beijing, China. [3]State Key Laboratory of Cell Biology, Shanghai Institute of Biochemistry and Cell Biology, Center for Excellence in Molecular Cell Science, Chinese Academy of Sciences, Shanghai 200031, China. [4]School of Life Science, Hangzhou Institute for Advanced Study, University of Chinese Academy of Sciences, Hangzhou 310024, China. [5]School of Chinese Materia Medica, Nanjing University of Chinese Medicine, Nanjing 210023, China. [6]School of Pharmaceutical Science and Technology, Hangzhou Institute for Advanced Study, University of Chinese Academy of Sciences, Hangzhou 310024, China. [7]Carbohydrate-Based Drug Research Center, CAS Key Laboratory of Receptor Research, State Key Laboratory of Drug Research, Shanghai Institute of Materia Medica, Chinese Academy of Sciences, Shanghai 201203, China. [8]Analytical Research Center for Organic and Biological Molecules, State Key Laboratory of Drug Research, Shanghai Institute of Materia Media, Chinese Academy of Sciences, Shanghai 201203, China. [9]School of Life Sciences, Fudan University, Shanghai 200433, China. [10]School of Life Science and Technology, Shanghai Tech University, Shanghai 200120, China. [11]These authors contributed equally: Lin-Lin Zhou, Tao Zhang, Yun Xue. ✉e-mail: hbji@sibcb.ac.cn; yangcg@simm.ac.cn

chemo-activation of *Hs*ClpP can also inhibit mitochondria OXPHOS to exert anticancer effects[5–13].

*Hs*ClpP is highly expressed in multiple tumor tissues and has been reported to be oncogenic in several types of cancers[14]. Lung squamous cell carcinoma (LUSC) accounts for approximately 30% of non-small cell lung cancer. And no significant difference in expression level of *CLPP* is observed between male and female patients according to the UALCAN dataset (http://ualcan.path.uab.edu/index.html)[15,16]. Chemotherapy and radiotherapy are the primary therapeutic options for LUSC, while targeted therapy has few benefits for LUSC patients, highlighting the need for effective target-based therapy for LUSC patients[17]. However, the role of *Hs*ClpP in LUSC development and the therapeutic effect of selective and potent chemo-activation of *Hs*ClpP is still not established and should be carefully explored.

Hyperactivating *Hs*ClpP through imipridones (e.g., ONC201 and ONC212) was reported to be a potential therapeutic strategy for treating acute myeloid leukemia (AML) in vivo recently[7,8]. Based on the scaffold of ONC201, ClpP activators such as 16z, TR-107 and IMP075 with enhanced enzymatic and anti-proliferative activity in cancer cells were developed. These derivatives also showed anticancer efficacy in vivo[9,10,18]. We also identified pyrimidone ZG111 as a different *Hs*ClpP activator with promising therapeutic potential for treating pancreatic ductal adenocarcinoma (PDAC)[11]. However, these activators are not specifically targeting to *Hs*ClpP since the amino acid sequence and protein folding are highly conserved in bacteria and eukaryotes. It is noteworthy that compound D9 was reported to behave as a selective activator of *Hs*ClpP rather than the bacterial ClpP proteins in vitro[19]. However, other researchers declared that D9 is a weak activator in vitro and in SUM159 tumor cells[7]. These data showed that the specie-specific activator for *Hs*ClpP with high cellular potency has not yet been identified. Non-selective activators that should, for example, target *Hs*ClpP for anticancer effect could also activate ClpPs of gut microbes, which might cause negative effects during cancer therapy. While the selective activation on *Staphylococcus aureus* ClpP (*Sa*ClpP) has been reported[20], achieving selective activation of *Hs*ClpP remains challenging but pivotal[6,21–23].

In this study, we develop a potent activator ZK53 that is highly selective on *Hs*ClpP but inactive toward bacterial ClpP proteins. ZK53 treatment allosterically activates *Hs*ClpP to uncontrollably degrade the essential mitochondrial proteins, such as ETC subunits, induces mitochondrial dysfunction, and causes cell cycle arrest, therefore showing anticancer effects on LUSC in vivo.

## Results

### Screening hit BX471 selectively activates *Hs*ClpP
Several representative activators nonspecifically activate both *Hs*ClpP and *Sa*ClpP in vitro, such as ADEP 4, ONC212, and ZG180 (Fig. 1a)[20], while the selectivity of the others has not been investigated and the use of species-specific and highly potent *Hs*ClpP activators for anticancer therapy have yet to be explored. To identify different chemical scaffolds of ClpP activators, we previously conducted high-throughput screening (HTS) of a small-molecule library containing 3,896 compounds using a fluorescence intensity (FI)-based protease activity assay[11]. One of the hits in the HTS, BX471, a selective antagonist of the CC chemokine receptor-1 (CCR1), exhibits potent activation on *Hs*ClpP (Fig. 1b)[24]. Then, we ran a PAGE-based assay, in which BX471 efficiently activates *Hs*ClpP with a quantified $EC_{50}$ of 6.07 μM (Fig. 1c). Interestingly, BX471 does not activate *Sa*ClpP for α-casein hydrolysis at a high concentration up to 80 μM in vitro (Supplementary Fig. 1a).

### Synthetic lead ZK53 shows improved activation on *Hs*ClpP and directly binds to *Hs*ClpP
To improve the potency of BX471 on the chemo-activation of *Hs*ClpP, we performed synthetic optimization. Considering the hydrophobic

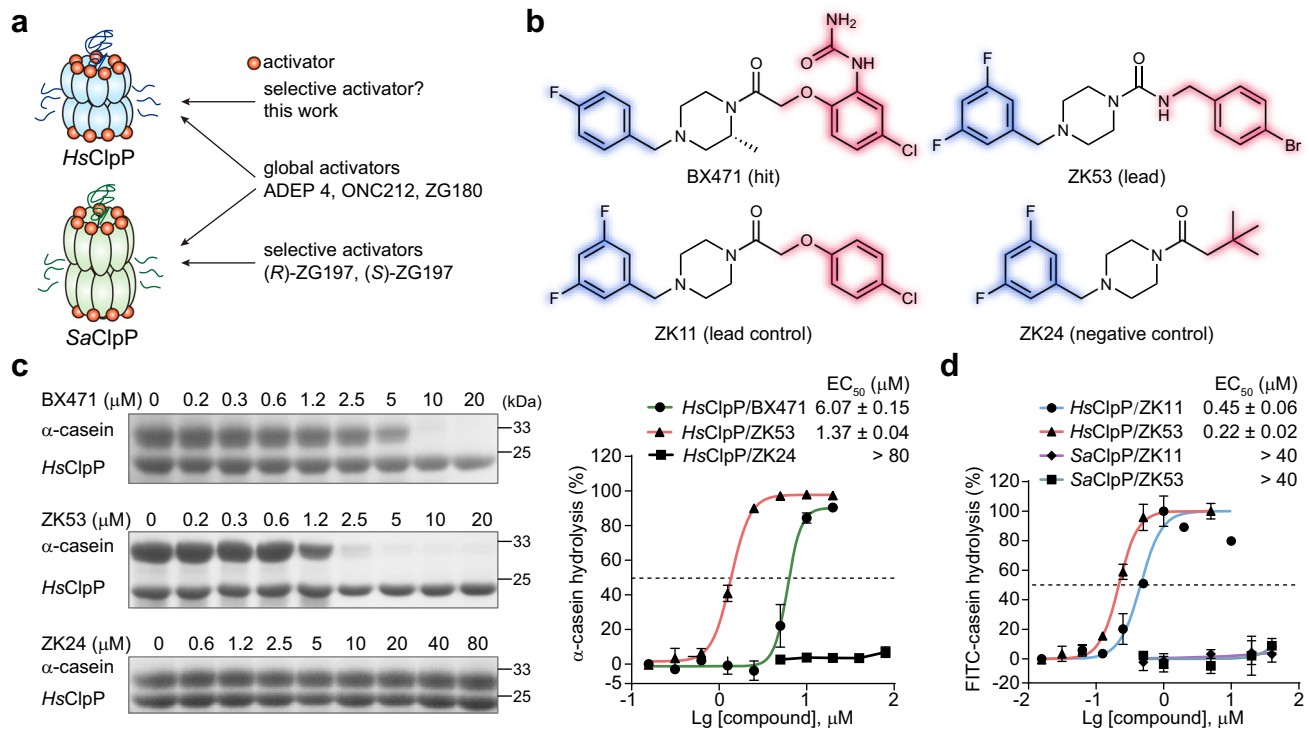

**Fig. 1 | ZK53 selectively activates *Hs*ClpP in vitro. a** Schematic summary of known activators for *Hs*ClpP and *Sa*ClpP. The goal of this work is to develop a selective activator of *Hs*ClpP. **b** Chemical structures of selective *Hs*ClpP activators. The screening hit BX471, two synthetic leads, ZK11 and ZK53, and the negative control ZK24 are shown. The fixed motif is highlighted in blue, and the substituted moieties are highlighted in red. **c** Effect of ClpP activators on α-casein hydrolysis by *Hs*ClpP in the PAGE-based assay. Representative gel (left) of three independent experiments and the quantitation of the $EC_{50}$ (right) are shown. **d** Quantitation of the effect of ZK11 and ZK53 on FITC-casein hydrolysis by *Hs*ClpP and *Sa*ClpP in the FI-based assay ($n = 3$ biological samples). All data are represented as mean ± SD (error bars).

property of ligand binding pockets in ClpP, we omitted the hydrophilic guanidyl group and also removed the methyl substituent in BX471 to simplify analogs synthesis. 3,5-difluorophenyl was introduced as a privileged motif to obtain ZK11 with increased activity (Fig. 1b). We further incorporated the 1-(4-bromobenzyl)urea fragment and yielded ZK53 as a lead compound that exhibits significantly improved activation on *Hs*ClpP compared to BX471. The $EC_{50}$ of ZK53 was determined to be 1.37 μM for α-casein hydrolysis by *Hs*ClpP (Fig. 1b, c), while ZK53 displays minimal activation on *Sa*ClpP as well as *Escherichia coli* ClpP (*Ec*ClpP) (Supplementary Fig. 1b). ZK24 was synthesized and used as a negative control with no activation on *Hs*ClpP and *Sa*ClpP (Fig. 1b, c, and Supplementary Fig. 1c). Additionally, ZK11 activates *Hs*ClpP with an $EC_{50}$ of 0.45 μM and ZK53 shows a two-fold stronger $EC_{50}$ of 0.22 μM in the Fl-based protease activity assay, whereas they are inactive on *Sa*ClpP (Fig. 1d). These data indicate that ZK11 and ZK53 are selective *Hs*ClpP activators with minimal effects on bacterial ClpP. Though the inhibition of CCR1 was not associated with anticancer[24], the in vitro inhibitory effect of ZK53 on CCR1 should be explored in future.

We next investigated the interactions of our activators and *Hs*ClpP in vitro. We first detected the binding affinity using the differential scanning fluorimetry (DSF) assay[25]. ZK11 and ZK53 significantly increase the melting temperature (Tm) of the recombinant *Hs*ClpP protein by 10.7 °C and 16.1 °C, respectively, when assayed at a molar ratio of 1:10, while ZK24 shows no effect on the shift of Tm (ΔTm) (Supplementary Fig. 1d). Consistent with the weak activation on *Sa*ClpP, our selective *Hs*ClpP activators have little effect on the thermal stability of *Sa*ClpP (Supplementary Fig. 1e). As such, ZK11 and ZK53 selectively bind and stabilize *Hs*ClpP rather than *Sa*ClpP in vitro.

## Structure of the ZK53/*Hs*ClpP complex reveals the mode of action

To understand the ligand binding mode, we resolved the X-ray crystal structure of the recombinant *Hs*ClpPΔ56 protein bound with ZK53 at a resolution of 1.9 Å (PDB: 8HGK) (Supplementary Table 1). Due to poorly ordered electron density, some residues either at the terminus or in specific loops were omitted, and some side chains were truncated in the model (Supplementary Table 2). *Hs*ClpP assembles into a tetradecameric state with fourteen ZK53 molecules binding in the fourteen hydrophobic pockets, far from the active sites (Fig. 2a). The representative electron density maps of α2 and α4, which locate at the periphery and core of *Hs*ClpP monomer, respectively, demonstrated the map quality (Supplementary Fig. 2a, b). Also, the electron density for ZK53 in situ in the binding site clearly show the ligand-binding area and the existence of ZK53 binding in *Hs*ClpP (Supplementary Fig. 2c and Fig. 2b). Structural alignment analysis on the apo *Hs*ClpP and ZK53/*Hs*ClpP complex yields a low Cα root mean square deviation (RMSD) value of approximately 0.31 Å for the ClpP monomer and also reveals a conformational change from the extended state of the apo *Hs*ClpP to the compact state upon ZK53 binding (Supplementary Fig. 2d). Inspection of the binding pocket shows that no obvious conformational change was observed for E82, whereas ZK53 binding leads to rotation of the side chains of L79, V148 and L170 (Supplementary Fig. 2e). ZK53 shares the similar mechanism with most other *Hs*ClpP activators[6,8,11], while the discovery of activator-bound *Hs*ClpP in the extended state can help further elucidate the mode of action for ClpP activation.

The interaction of ZK53 and *Hs*ClpP is strongly stabilized by a π-π stacking effect between the 3,5-difluorobenzyl motif in ZK53 and the side chain of W146 in *Hs*ClpP. Extensive hydrophobic effects are observed between ZK53 with the adjacent residues of V148 and L170, and those between the 4-bromobenzyl motif in the ligand and the alkyl chains of I84, L79, and E82 in protein (Fig. 2c). Additionally, the *N*-atom connected to the difluorobenzyl group forms a hydrogen bond with the side chain of Y118 (Fig. 2c). Likewise, another *N*-atom that is

connected to the 4-bromobenzyl fragment interacts with Q107 via a water-mediated hydrogen bond.

## Mechanism of ZK53's selective activation on *Hs*ClpP other than bacterial ClpP

The crystal structure of the ZK53/*Hs*ClpP complex reveals that the W146 residue is essential for the selective binding of ZK53 with *Hs*ClpP via a π-π stacking effect, whereas the corresponding I91 in *Sa*ClpP unlikely provides a similar effect, thus weakening the binding of ZK53 to *Sa*ClpP (Fig. 2d). To verify these structural observations, we substituted W146 in *Hs*ClpP with I146 and also constructed an I91W mutant of *Sa*ClpP. The activation of W146I *Hs*ClpP by ZK11 and ZK53 was substantially weakened as evidenced by the increased $EC_{50}$ from 1.6 to 42.4 μM for ZK11 and from 1.4 to 11.8 μM for ZK53, while activation of I91W *Sa*ClpP was significantly enhanced with the decreased $EC_{50}$ values from over 80 to 8.4 μM for ZK11 and from 44.5 to 5.4 μM for ZK53 (Fig. 2e and Supplementary Fig. 2f, g). We wondered whether the declined activation of W146I *Hs*ClpP and the elevated activation of I91W *Sa*ClpP are due to changes in the binding affinity between our activators and the target proteases. So, we performed a DSF assay to record activator-induced ΔTm of the ClpP variants. Indeed, the ΔTm of W146I *Hs*ClpP dramatically decreased in the presence of ZK11 and ZK53 compared to that of *Hs*ClpP, while the ΔTm of I91W *Sa*ClpP increased compared to that of *Sa*ClpP (Fig. 2f and Supplementary Fig. 2h). These data reveal that the π-π stacking effect between W146 and the difluorobenzyl moiety is required for our activators to achieve high selectivity on *Hs*ClpP but not *Sa*ClpP, and a single amino acid substitution can switch the binding affinity and activation efficacy on the two proteins.

The π-π stacking effect seems to be a discriminator when the activator has a simple central scaffold, while the global activator ADEP 4, bearing a rigid scaffold, displays no difference in binding and activation on both ClpP proteins and their mutants (Fig. 2e, f and Supplementary Fig. 2h, g). Other global activators, such as ONC212, are similarly featured on rigid scaffolds, and the π-π stacking effect is not essential for activator interactions (Supplementary Fig. 2i). Collectively, these data demonstrate that the selective activation of ZK53 on *Hs*ClpP primarily relies on the hydrophobic π-π stacking effect between the difluorobenzyl moiety in the ligand and W146 residue in *Hs*ClpP.

Given that Y118 is a peripheral part of the ligand-binding pocket, it would seem relevant to understand whether the Y118A ClpP mutant, an uncontrollable gain-of-function variant, still binds ZK53 and if this is still effective. Similar with previous findings, Y118A ClpP functions as a self-activated protease that is capable of degrading α-casein protein in vitro (Supplementary Fig. 2j)[8,11]. Interestingly, ZK53 is still effective to promote Y118A ClpP for α-casein degradation in PAGE assay (Supplementary Fig. 2k). To test the thermal stabilization effects of ClpP activators on the Y118A ClpP mutant, we performed the NanoDSF assay and found that most activators, except ADEP 4, still showed comparable thermally stabilized effects on ClpP WT and Y118A mutant in vitro (Supplementary Fig. 2l), which indicates that Y118 is not required for ZK53 binding and activating *Hs*ClpP.

We then investigated whether our activators activate ClpPs in gut microbes. We purified the recombinant *Limosilactobacillus reuteri* ClpP (*Lr*ClpP) and found that ZK11 and ZK53 barely activate *Lr*ClpP, while low concentrations of ADEP 4 and ONC212 substantially activate *Lr*ClpP for α-casein hydrolysis (Supplementary Fig. 2m). Consistently, ZK11 and ZK53 minimally inhibit the growth of human gut microbe strains, such as *Limosilactobacillus reuteri* (*LR*), *Lactobacillus rhamnosus GG* (*LGG*) and *Bifidobacterium longum subsp. infantis* (*BL*), while the global activator ADEP 4 has significant inhibitory effects on the growth of these strains (Fig. 2g and Supplementary Fig. 2n). ONC201 also inhibits the growth of gut microbes at high concentration. These data indicate that selective *Hs*ClpP activators can relief side-effects during

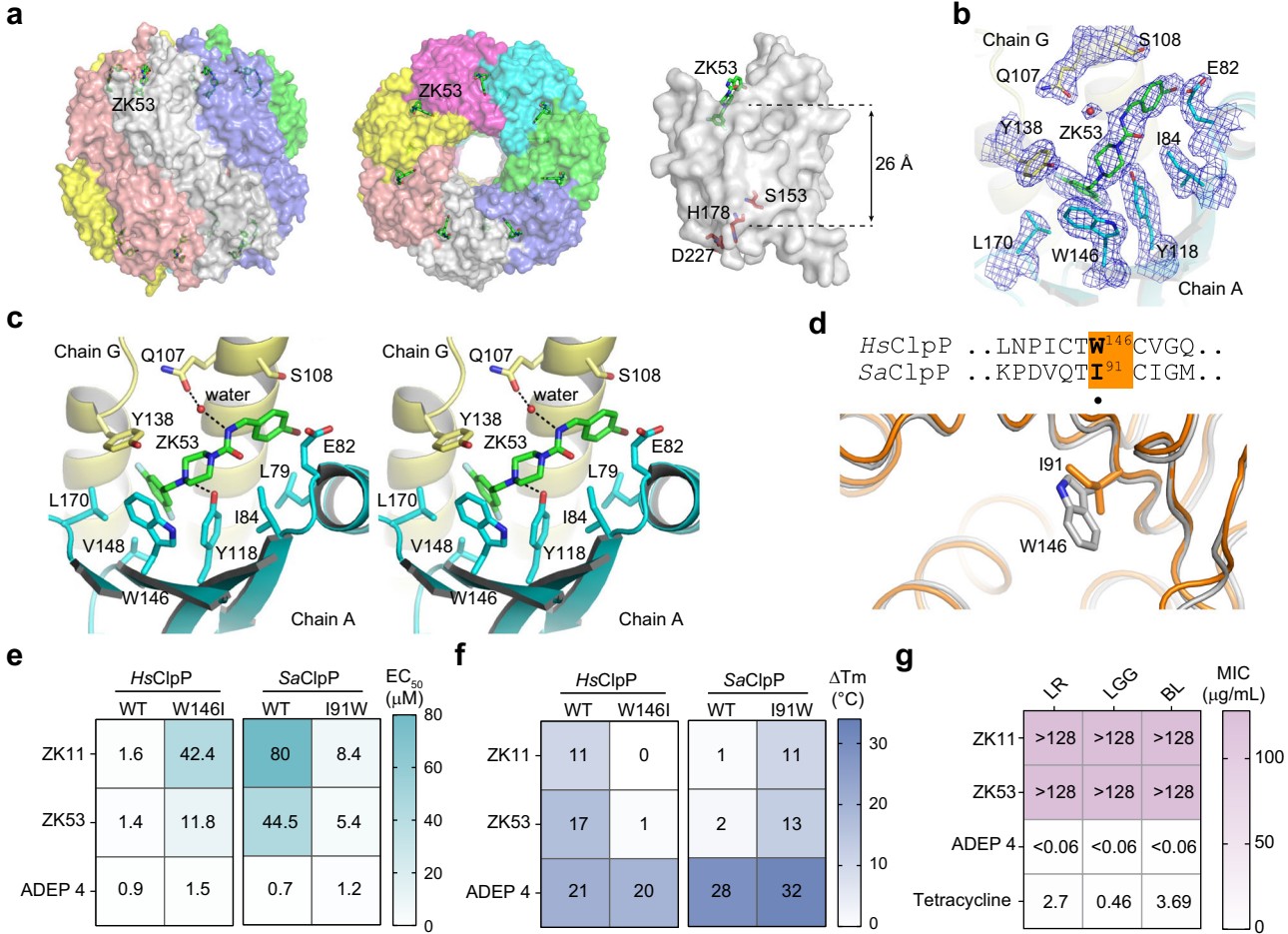

**Fig. 2 | Mechanism of the selective interaction of ZK53 and *Hs*ClpP. a** Overall structure of ZK53/*Hs*ClpP complex, shown from the side (left) view and top view (middle), and the active sites in the monomer of *Hs*ClpP (right). *Hs*ClpP protein is shown in surface, ZK53 is shown in green sticks, and the residues of active sites are shown in red sticks. Structural drawings for ZK53/*Hs*ClpP complex were performed in PyMOL. **b** 2fo-fc density map contoured to 1.2σ (blue mesh) for ZK53 in situ binding in the hydrophobic site. The amino acids in chain G are presented as yellow sticks and those in chain A are shown as cyan sticks. **c** Stereo-view of the interactions of ZK53 binding to *Hs*ClpP. The key amino acids involved in interactions with ZK53 are shown by sticks. The hydrogen bonds are indicated in dark dotted lines.

**d** Part of the sequence (top) and structure (bottom) alignments of *Hs*ClpP and *Sa*ClpP showing the essential residues for selective binding with activators. These amino acids are highlighted with an orange background in the sequence alignment. The side chains of the two amino acids are shown in stick. **e** Heatmap showing the $EC_{50}$ values of activators-promoted α-casein hydrolysis by ClpP variants in the PAGE-based assay. **f** Heatmap representation of ΔTm values showing the activators' effect on the thermal stabilization of ClpP variants. **g** Antibiotic effects of ClpP activators and tetracycline on gut microbes. Tetracycline was assayed as a positive control. The $IC_{50}$ is displayed in a heatmap.

## The gain-of-function Y118A *Hs*ClpP mutant impairs LUSC cells proliferation and inhibits tumor growth in vivo

Generally, aberrant activation of *Hs*ClpP inhibits tumorigenesis of leukemia and some solid tumors[8–11], which prompts us to ask whether gain-of-function *Hs*ClpP similarly exhibits anticancer effects in LUSC. The Y118A *Hs*ClpP mutant degrades substrate proteins independently on activators, and exhibits anticancer effects in vivo[11,26]. Thus, we transduced the doxycycline (Dox)-inducible Y118A *Hs*ClpP lentiviruses into LUSC cells and evaluated the long-term effects of *Hs*ClpP activation in the clonogenic assay. As expected, the Dox-induced overexpression of Y118A *Hs*ClpP hinders the clonogenic capability of LUSC cells during a ten-day treatment (Supplementary Fig. 3a). In addition, Y118A *Hs*ClpP overexpression significantly inhibits the cell proliferation of the SK-MES-1, H226, and H1703 LUSC cell lines (Supplementary Fig. 3b), and suppresses tumor growth of xeno-transplanted H1703 cells in Balb/c nude mice (Fig. 3a, b). *Hs*ClpP functions directly in degrading misfolded and damaged proteins in mitochondria;

therefore, aberrant activation of *Hs*ClpP could cause heavy degradation of multiple mitochondrial proteins, including ETC subunits[3]. Consistently, Y118A *Hs*ClpP overexpression downregulates the abundance of several ETC subunits in SK-MES-1 and H1703 cell lines (Fig. 3c).

We also explored *CLPP* mRNA levels in LUSC patients in the UALCAN dataset (http://ualcan.path.uab.edu/index.html)[15,16]. This analysis shows a significantly higher transcript level of *CLPP* in primary tumors than in normal tissues, which indicates a safety window for small-molecule activators selectively targeting *Hs*ClpP in cancer cells (Fig. 3d). Therefore, these data demonstrate that aberrantly activating *Hs*ClpP inhibits LUSC cell proliferation and tumorigenesis, making it a promising therapeutic strategy for LUSC treatment.

## *Hs*ClpP activators inhibit LUSC cell proliferation

Next, we assessed whether our *Hs*ClpP activators show a similar anticancer effect as the Y118A *Hs*ClpP overexpression on LUSC cell lines. ZK11 and ZK53 inhibit LUSC cell viability in a dose-dependent manner in the MTT assay, while ZK53 exhibits a four-fold stronger inhibitory effect than ZK11, as evidenced by the measured growth

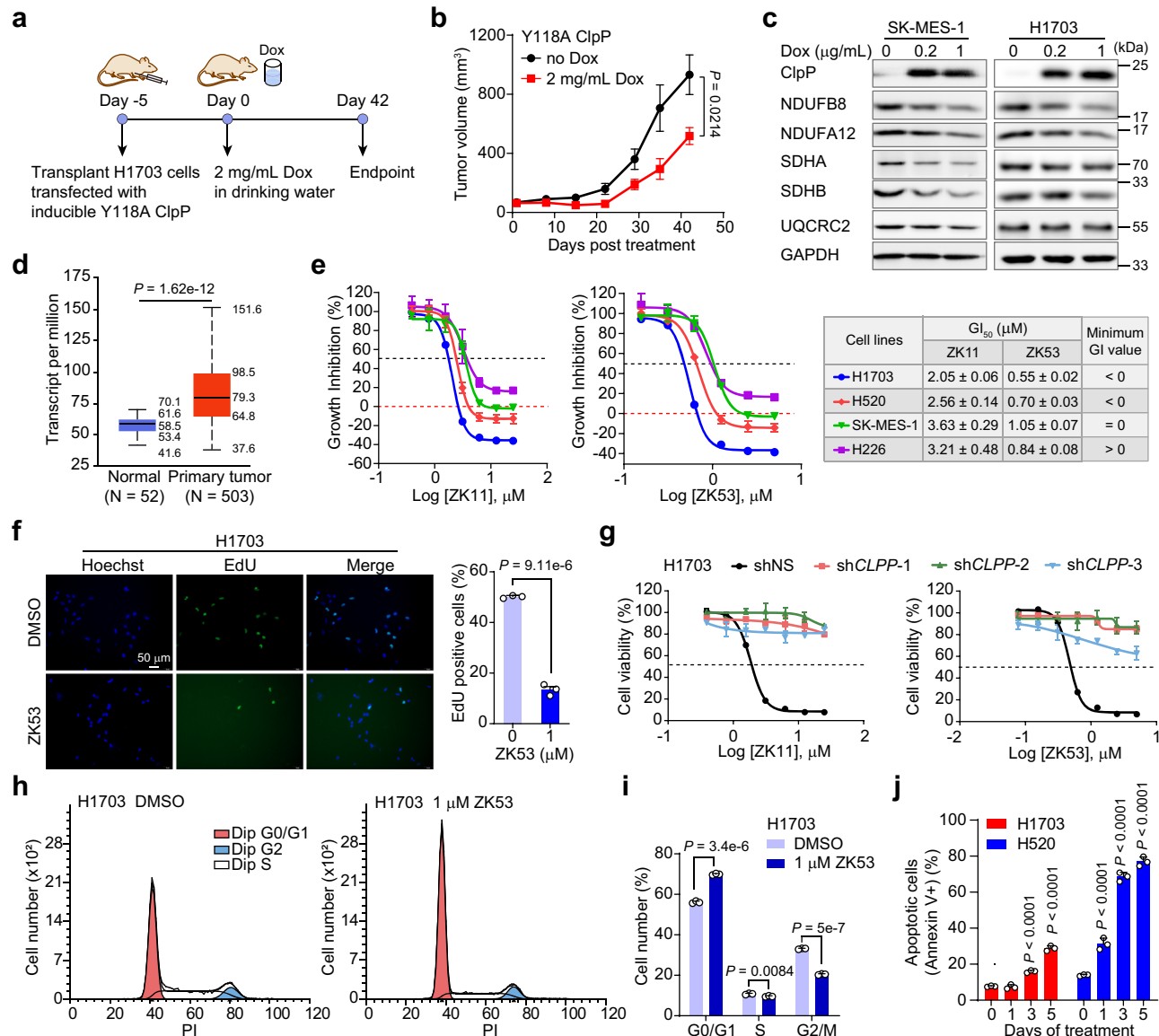

**Fig. 3 | The effects of *Hs*ClpP activation on lung squamous carcinoma (LUSC) cell proliferation. a** Schematic illustration of the xenograft mice models transplanted with H1703 cells that overexpress Y118A ClpP by doxycycline (Dox)-induction. **b** Effect of Y118A ClpP overexpression on tumor growth of H1703 cells in xenograft mice. Female nude mice (*n* = 5 animals) were fed without or with Dox in drinking water. The *P* values were calculated by a two-sided Student's *t* test with confidence interval of 95%. **c** Effect of Y118A ClpP overexpression on the protein abundance of ETC subunits in SK-MES-1 and H1703 cell lines. The images shown are representative of three independent experiments. **d** The mRNA level of *CLPP* in LUSC patients from the UALCAN database. The maxima, Q3, median, Q1, and minima of each group have been displayed from top to bottom in the graph. Statistical differences were analyzed using a two-tailed unpaired Student's *t* test. **e** Inhibitory effect of ZK11 and ZK53 on the cell viability of LUSC cell lines. The growth inhibition (GI)$_{50}$ values and minimum GI values are listed in the table (*n* = 3 biological samples). **f** Representative images showing the inhibitory effect of ZK53 on H1703 cells in the EdU proliferation assay (*n* = 3 biological samples). The percentage of EdU positive (green) cells to total cell count (blue) is calculated. Scale bar, 50 μm. The *P* values were calculated by a two-sided Student's *t* test with confidence interval of 95%. **g** Effect of ZK11 and ZK53 on the viability of shNS and sh*CLPP* H1703 cells (*n* = 3 biological samples). shNS, the control shRNA; sh*CLPP*, shRNA for *CLPP* knockdown. Error bars represent the mean ± SD. Effect of ZK53 on cell cycle of H1703 cells (**h**) and the quantitation graph (**i**) (*n* = 3 biological samples). The *P* values were calculated by a two-sided Student's *t* test with confidence interval of 95%. **j** Apoptosis analysis of H1703 and H520 cells treated with 1 μM ZK53 for the indicated time points (*n* = 3 biological samples). The *P* values were calculated by one-way ANOVA with Dunnett's multiple comparisons test. All data are represented as mean ± SD (error bars).

inhibition (GI)$_{50}$ values (Fig. 3e)[27]. The minimum GI value is below zero for H1703 and H520 cell lines (Fig. 3e), which reveals a cytotoxic effect of ZK11 and ZK53; is equal to zero for SK-MES-1 cells, which indicates a cytostatic effect; and is above zero for H226 cells, which indicates that our activators cannot completely prevent H226 cell proliferation. Our selective *Hs*ClpP activators also inhibit cell proliferation in the clonogenic assay (Supplementary Fig. 3c) and the time-dependent MTT assay of LUSC cells (Supplementary Fig. 3d). To visualize the effect of ZK53 on the proliferation of living

cells, we directly detected the efficacy of DNA synthesis when LUSC cells were exposed to ZK53 in the EdU incorporation assay. This shows that ZK53 treatment significantly inhibits the proliferation of LUSC cells (Fig. 3f and Supplementary Fig. 3e). Additionally, we tested the inhibitory effects of ZK53 on the viability of non-malignant cells, such as the human lung fibroblast MRC-5 cells. ZK53 shows a three-fold higher IC$_{50}$ value in MRC-5 cells compared with the LUSC H1703 cells (Supplementary Fig. 3f). Moreover, ZK53 does not induce cell apoptosis in MRC-5 cells (Supplementary

Fig. 3g). These data suggest low cytotoxicity of ZK53 in normal lung tissues.

Next, we used the genetic approach to determine whether the protein abundance of *Hs*ClpP could influence the sensitivity of LUSC cells to our activators. Knockdown of *CLPP* significantly weakens the inhibitory effects of ZK11 and ZK53 on cell viability in H1703 and SK-MES-1 cell lines (Fig. 3g and Supplementary Fig. 3h, i). We also constructed LUSC cell lines overexpressing *CLPP* to test drug response (Supplementary Fig. 3j). Contrary to the *CLPP* knockdown cells, ZK11 and ZK53 show a more substantial inhibitory effect on H1703 and SK-MES-1 cells overexpressing *CLPP* (Supplementary Fig. 3k, l). Collectively, these data demonstrate that our *Hs*ClpP activators inhibit the viability of LUSC cells, which depends on the abundance of *Hs*ClpP.

Consistent with the strong inhibitory effect on cell proliferation, ZK53 arrests cell-cycle progression at the G0/G1 phase in H1703 and H520 cell lines during a cell cycle analysis (Fig. 3h, i and Supplementary Fig. 3m, n). ZK53 also induces apoptosis in H1703, H520 and SK-MES-1 cell lines (Fig. 3j and Supplementary Fig. 3o). These data demonstrate a strong anti-proliferative effect of *Hs*ClpP activator ZK53 on LUSC cells.

## ZK53 hampers mitochondrial functions and targets *Hs*ClpP in LUSC cells

Given that the chemo-activation of *Hs*ClpP can disturb the homeostasis of mitochondrial proteins, such as ETC subunits, mitochondrial transcription factor A (TFAM), the mitochondrial protein elongation factor Tu (TUFM), and TCA cycle enzyme aconitase (ACO2)[7–9,11], we assessed the effects of ZK53 on these proteins in LUSC cells. ZK53 triggers a substantial reduction in multiple ETC subunits, including NDUFB8, SDHA, UQCRC2, and COXIV in a dose-dependent manner (Fig. 4a and Supplementary Fig. 4a). ZK53 also reduces the protein abundances of TFAM, TUFM, and ACO2 in H1703 cells as ONC201 (Supplementary Fig. 4a). Knockdown of *CLPP* prevents ZK53-induced uncontrolled degradation of mitochondrial proteins, while overexpression of *CLPP* enhances this phenomenon (Fig. 4b and Supplementary Fig. 4b), indicating the *Hs*ClpP-dependent degradation of mitochondrial proteins by ZK53 in LUSC cells.

As a result of impaired ETC subunits, the OXPHOS capability is largely disturbed in LUSC cells by monitoring the oxygen consumption rate (OCR) in Seahorse energy analysis (Fig. 4c and

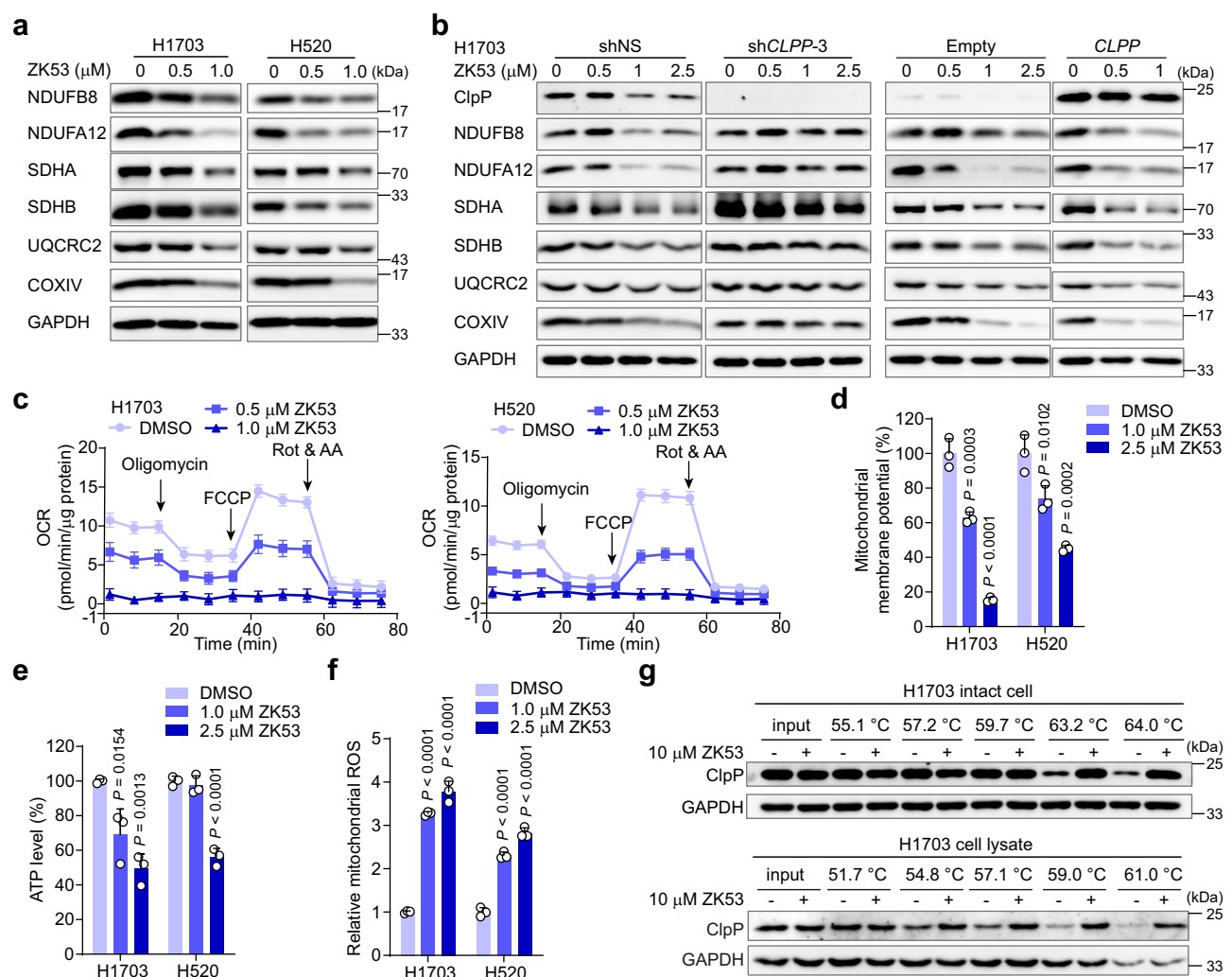

**Fig. 4 | ZK53 impairs mitochondrial functions and targets *Hs*ClpP in LUSC cells.**
**a** Effect of ZK53 on protein abundance of ETC subunits in H1703 and H520 cell lines. The images shown are representative of three independent experiments.
**b** Immunoblot showing the effect of ZK53 on the protein abundance of ETC subunits in H1703 cells with *CLPP* knockdown (left two columns) or overexpression (right two columns). The images shown are representative of three independent experiments. **c** Effect of ZK53 on the oxygen consumption rate (OCR) in H1703 and H520 cells using Seahorse Analyzer (*n* = 6 biological samples). Rot rotenone, AA

antimycin A. Effect of ZK53 on the mitochondrial membrane potential (**d**), ATP level (**e**), and mitochondrial ROS (**f**) in H1703 and H520 cell lines (*n* = 3 biological samples). The *P* values are calculated by one-way ANOVA with Dunnett's multiple comparisons test. **g** Effect of ZK53 on the thermal stability of *Hs*ClpP protein in intact cell and cell lysate of H1703 cell line in CETSA. The images shown are representative of three independent experiments. All data are represented as mean ± SD (error bars).

Supplementary Fig. 4c). Since the mitochondrial membrane potential (MMP) is an indicator for electron transport and OXPHOS, we observed significantly decreased MMP in LUSC cells upon ZK53 treatment using the potential-sensitive probe JC-1 (Fig. 4d and Supplementary Fig. 4d). The MMP is believed to be the driving force for ATP production. As a result of the decreased MMP and impaired OXPHOS, ATP production is significantly decreased in H1703 and H520 cells upon ZK53 treatment (Fig. 4e), while the ATP levels are unaffected in MRC-5 cells in the presence of ZK53 (Supplementary Fig. 4e). Mitochondria are the main generators of reactive oxygen species (ROS)[28]. So, we tested the effects of ZK53 treatment on mitochondrial ROS and observed a significant increase in LUSC cells (Fig. 4f and Supplementary Fig. 4f). The mtDNA is similarly decreased in H1703 and SK-MES-1 cells after ZK53 treatment (Supplementary Fig. 4g). Collectively, ZK53 treatment results in severe dysregulation of mitochondrial functions in LUSC cells.

Furthermore, we assessed whether our activators directly interact with *Hs*ClpP in LUSC cells by running the cellular thermal shift assay (CETSA). ZK53 dramatically increases the thermal stability of *Hs*ClpP in both intact cell and cell lysate of H1703 cells, indicating a direct ligand binding to the target protein (Fig. 4g)[29]. On the contrary, our activators ZK11 and ZK53 barely influence the thermal stability of *Sa*ClpP, while the global activator ADEP 4 does (Supplementary Fig. 4h). These data demonstrate that ZK53 directly targets the mitochondrial ClpP protein in LUSC cells.

## ZK53 inhibits the adenoviral early region 2 binding factor (E2F) targets in H1703 cells

To better understand the molecular mechanisms underlying the anti-proliferative effects of ZK53 on LUSC cell lines, we conducted transcriptome-wide RNA-sequencing analysis (RNA-seq) of H1703 cells after treatment with 1 μM ZK53. Approximately 1,990 genes were calculated as differentially expressed genes (DEGs) (Supplementary Fig. 5a). Then, KEGG enrichment analysis was performed to functionally annotate the DEGs. In line with the observed inhibitory effects on the LUSC cell cycle in flow cytometry (Fig. 3h, i), the cell cycle pathway is the most significantly enriched downregulated pathway in both KEGG and Reactome enrichment analysis (Fig. 5a and Supplementary Fig. 5b). Gene set enrichment analysis (GSEA) showed that ZK53 treatment significantly downregulates the E2F targets (Fig. 5b). E2F targets are also observed as the top-rank gene set in the gene set variation analysis (GSVA) (Fig. 5c and Supplementary Fig. 5c).

E2F is the major transcriptional factor responsible for the G1/S transition, which transactivates the essential target genes for entering the S phase, while the transcriptional suppression of E2F target genes impedes the G1/S phase transition and causes cell cycle arrest[30]. We first performed qPCR analysis and confirmed that the transcription of several E2F target genes, including *CCNA2*, *CCNE2*, *PCNA*, *MCM3*, and *MCM6*, are dose-dependently decreased in the presence of ZK53 in H1703 cells (Fig. 5d). The hyperphosphorylated retinoblastoma protein (Rb) directly binds to E2F and blocks its transcriptional activity. E2F is released and exhibits transcriptional activity when the cyclin D-CDK4/6 and cyclin E-CDK2 complexes consecutively phosphorylate the Rb protein (Supplementary Fig. 5d)[31]. We next examined the Rb phosphorylation and the levels of those proteins responsible for Rb phosphorylation using immunoblot. As expected, ZK53 treatment results in a dramatic decrease in S795, T826, and S807/811 phosphorylation on Rb in H1703 cells (Fig. 5e), which is partially responsible for the decrease in E2F target genes. We also observed the decreased protein levels of cyclin D1, CDK2, and cyclin E2 in H1703 cells upon ZK53 treatment, which could lead to the deficient phosphorylation of Rb (Fig. 5e). Collectively, the exposure of LUSC cells to ZK53 impair the essential regulatory pathways for the G1/S transition.

## ZK53 activates the ataxia-telangiectasia mutated (ATM)-mediated DNA damage response in H1703 cells

Considering that the decreased level of cyclin D1 was reported to be associated with DNA damage[32], we wondered whether ZK53-induced downregulation of cyclin D1 could be linked to DNA damage in LUSC cells. We then determined the phosphorylated histone H2AX (γ-H2AX) marker to reflect the DNA double-strand breaks (DSB)[33]. Immunofluorescence showed elevated γ-H2AX levels in H1703 cells upon treatment of 1 μM ZK53 (Supplementary Fig. 5e). To further visualize the DSB at the single cell level, we performed the alkaline comet assay[34], which shows an increase in the tail moment, representing increased DNA damage (Fig. 5f). It has been reported that ATM, an essential regulatory kinase, functions in the DSB-activated DNA damage response (DDR) signaling (Supplementary Fig. 5f)[35]. Indeed, we observed increased phosphorylation of p-ATM (S1981) in H1703 cells upon ZK53 treatment in immunoblot (Fig. 5g). As a result of ATM activation, elevated p-CHK2 (T68) and γ-H2AX, and declined CDC25A are observed (Fig. 5g and Supplementary Fig. 5f)[35–38].

We used the ATM-specific inhibitor KU-55933 to investigate the role of ATM activation in ZK53-treated H1703 cells[39]. KU-55933 treatment inhibits the elevated phosphorylation of p-ATM resulted from ZK53 treatment (Fig. 5h). Meanwhile, KU-55933 treatment also decreases p-CHK2 and γ-H2AX, the two substrates of ATM, indicating that the ZK53-induced increased p-CHK2 and γ-H2AX are primarily mediated through the ATM signaling pathway (Fig. 5h). Furthermore, both ATM inhibitors of KU-55933 and AZD1390 significantly mitigate the anti-proliferative effects of ZK53 on H1703 cells (Fig. 5i)[40], which suggests that ZK53 mainly inhibits cell proliferation through ATM activation. We conclude that ZK53 treatment activates the ATM-mediated DDR and induces anti-proliferation in H1703 cells.

We also performed proteomic comparisons with ONC201 in LUSC cells. We assessed the effects of ONC201 on the ZK53-regulated pathways in H1703 cells and found that both ZK53 and ONC201 downregulate the Rb phosphorylation and the regulatory proteins required for Rb phosphorylation, activate the DDR pathway, and inhibit the E2F targets transcript level in LUSC cells (Supplementary Fig. 5g, h). We also found ZK53 similarly regulates the reported ONC201-activated pathways in LUSC cells, including Akt-Erk, ATF4, and DR5 (Supplementary Fig. 5i)[41,42]. Like chemo-activation of *Hs*ClpP, the Y118A ClpP mutant exhibits similar outcomes in H1703 and SK-MES-1 cell lines (Supplementary Fig. 5j). Collectively, these data show that the two quite different druglike molecules, ZK53 and ONC201 regulate similar cell pathways in response to mitochondrial dysfunctions in LUSC cells.

## ZK53 exhibits anticancer activity in xenograft and autochthonous mouse models

Lastly, we evaluated the therapeutic effects of ZK53 on the in vivo mice model. The Balb/c nude mice were inoculated with H1703 cells and intraperitoneally administered ZK53. ZK53 treatment efficiently inhibits the tumor growth of H1703 cells in the xenograft mouse model (Fig. 6a). We observed much smaller tumor volume and weight in ZK53-treated group than those in vehicle control at the endpoint of treatment (Fig. 6b, c and Supplementary Fig. 6a, b). The proliferating LUSC cells in the tumor are significantly inhibited by ZK53 treatment, as indicated by Ki67 staining (Fig. 6d). Only a slight decrease in body weight is observed in ZK53-treated groups compared to the vehicle control (Supplementary Fig. 6c). The major organs in the endpoint of treatment were subjected to hematoxylin and eosin (H&E) staining, and no noticeable pathological changes are observed between the vehicle and ZK53-treated mice (Supplementary Fig. 6d). We also harvested the blood for complete blood count analysis and plasma biochemical analysis, and the results showed no significant differences between the two groups (Supplementary Tables 3 and 4).

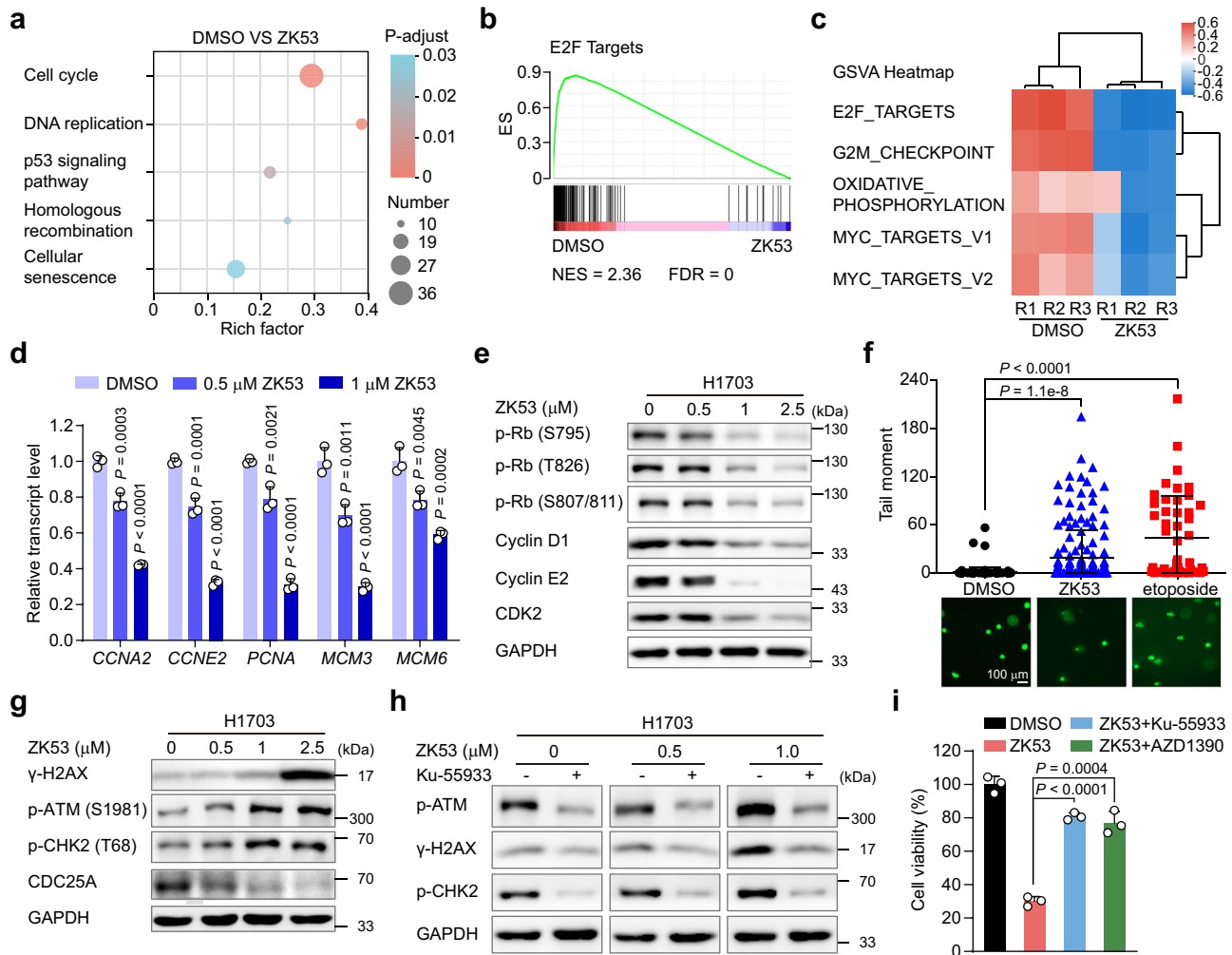

**Fig. 5 | ZK53 suppresses E2F targets and activates the ATM-mediated DDR.**
**a** KEGG enrichment analysis showing the top 5 enriched signaling pathways in ZK53-treated H1703 cells. The *P* values were calculated by a Fisher's exact test, and adjusted using the BH(FDR) method. **b** Gene set enrichment analysis (GSEA) reveals the E2F targets gene set as the most enriched hallmark gene set. NES, normalized enrichment score; FDR, false discovery rate. **c** Heatmap of gene set variation analysis (GSVA) of the differentially expressed genes (DEGs) in the ZK53-treated H1703 cells. The E2F targets gene set ranks at the top. **d** Effect of ZK53 on gene expression of E2F targets in H1703 cells measured by qPCR (*n* = 3 biological samples). The *P* values are calculated by one-way ANOVA with Dunnett's multiple comparisons test. **e** Effect of ZK53 on Rb phosphorylation and regulatory proteins involved in the cell cycle in H1703 cells in immunoblot analysis. The images shown are representative of three independent experiments. **f** Effect of ZK53 on DNA damage of H1703 cells in the alkaline comet assay. The results are processed by OpenComet (*n* = 145 images for DMSO, *n* = 170 images for ZK53, and *n* = 47 images for etoposide). Representative images are shown below the quantification graph. The *P* values were calculated by a two-sided Student's *t* test with confidence interval of 95%. Scale bar, 100 μm. **g** Effect of ZK53 on the DDR pathway in H1703 cells by immunoblot analysis. The images shown are representative of three independent experiments. **h** Effect of the specific ATM inhibitor, KU-55933, on the ZK53-induced DDR pathway in H1703 cells by immunoblot analysis. The images shown are representative of three independent experiments. **i** Effect of KU-55933 and AZD1390 each on ZK53-induced antiproliferation in H1703 cells (*n* = 3 biological samples). The *P* values were calculated by a two-sided Student's *t* test with confidence interval of 95%. All data are represented as mean ± SD (error bars).

To show the clinical relevance of our study, we evaluated the anticancer effects of ZK53 on the genetically engineered mouse model (Fig. 6e). Inactivating mutations of LKB1 are significantly concurrent with KRAS mutation, and the *Kras^{LSL-G12D/+};Lkb1^{fl/fl}* (KL) mouse model is established for the study of LUSC[43–45]. Both LUSC and lung adenocarcinoma (LUAD) were induced in our model as distinguished by histopathological analyses and immunostaining analyses of the LUSC biomarkers p40, SOX2, and KRT5 and the LUAD biomarker TTF1 (Fig. 6f and Supplementary Fig. 6e)[46]. We found that ZK53 treatment significantly suppressed the progression of these mouse LUSC, while minimally inhibited the development of mouse LUAD (Fig. 6f, g). The Ki67 immunostaining results further showed that LUSC cell proliferation was noticeably suppressed in ZK53-treated group (Fig. 6h, i), whereas we observed no significant loss of body weights in the ZK53-treated group compared with vehicle control (Supplementary Fig. 6f).

These results highlight the potential therapeutic effects of *Hs*ClpP activator ZK53 against tumor growth of LUSC cells in vivo.

## Discussion

We report on a highly selective and potent *Hs*ClpP activator, ZK53, showing anticancer effects in LUSC, which was not developed before our study. Targeting mitochondrial ETC has been increasingly investigated as a strategy for intervening in mitochondrial metabolism for cancer therapy[47]. ETC dysfunction not only perturbs the bioenergetic processes causing energy depletion but also reduces the biosynthesis of macromolecules, which collectively contributes to cell cycle arrest and apoptosis[48]. Different from the primary approach of direct inhibition on ETC complexes with small-molecule inhibitors, we demonstrated that ZK53-activated *Hs*ClpP degrades the ETC subunits and impairs the OXPHOS in LUSC cells. However, a recent study reported

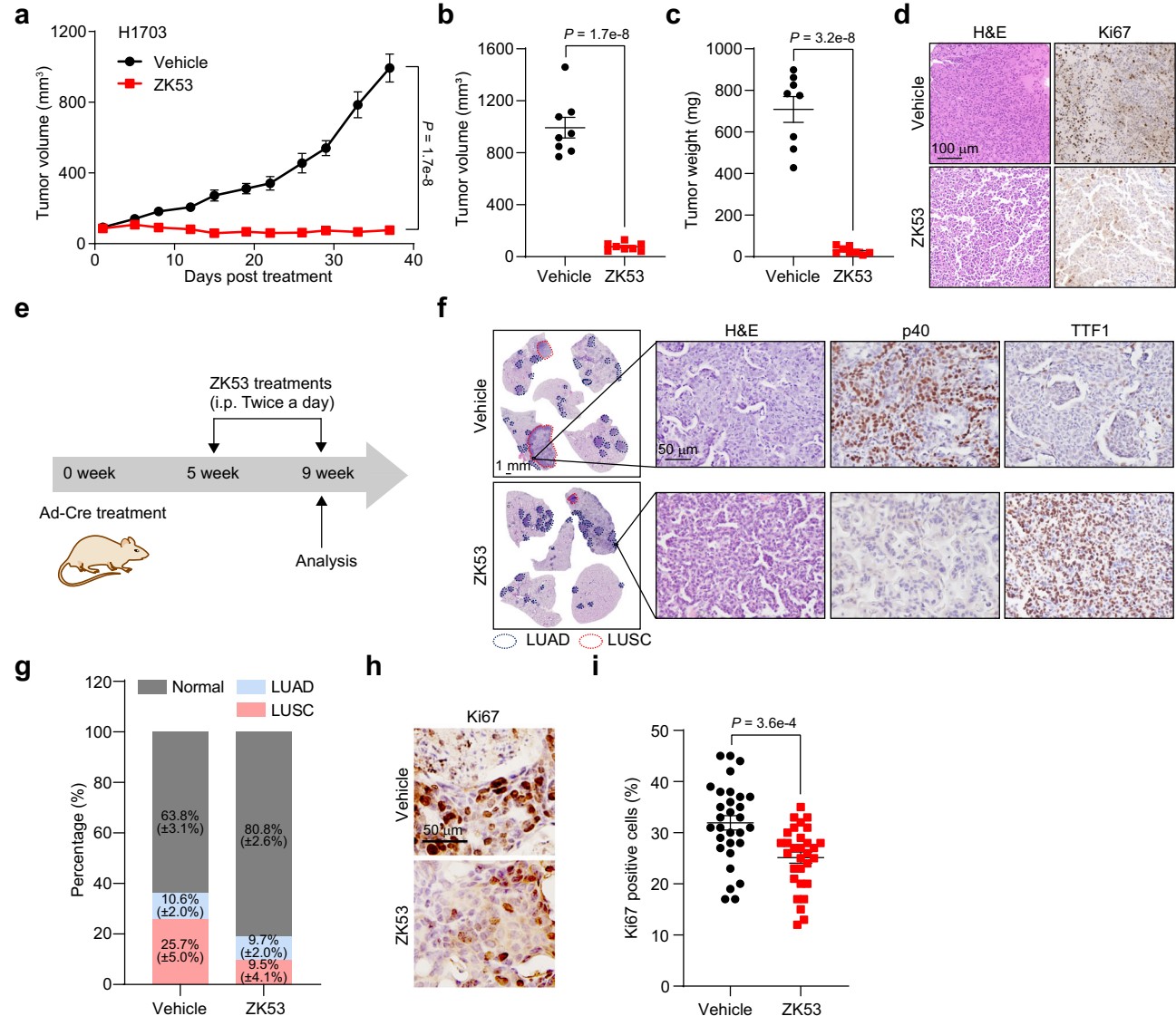

**Fig. 6 | ZK53 exhibits anticancer activity in vivo. a** Therapeutic effect of ZK53 on the tumor growth of H1703 cells in Balb/c nude mice ($n$ = 8 animals). The tumor volume (**b**) and tumor weight (**c**) on the day of euthanization ($n$ = 8 animals). **d** Representative images of hematoxylin and eosin (H&E) staining and Ki67 immunohistochemistry of LUSC tumor tissues. The image shown is a representative of three tests. Scale bar, 100 μm. **e** Schematic illustration of ZK53 treatment in KL mouse model. **f** Representative H&E staining and immunohistochemistry staining for TTF1 and p40 in LUAD and LUSC from KL mice treated with vehicle or ZK53. LUAD is indicated by blue dashed line, and LUSC is presented by red dashed line. Scale bar for whole lung: 1 mm; Scale bar for others: 50 μm. **g** Quantification of ratios of LUAD, LUSC, and normal tissue in whole lungs in KL mice treated with vehicle or ZK53 ($n$ = 6 animals). Representative Ki67 immunostaining (**h**) and statistical analyses (**i**) ($n$ = 30 images). Scale bar, 50 μm. The $P$ values were calculated by a two-sided Student's $t$ test with confidence interval of 95%. Figure 6i is represented as mean ± SD (error bars), and all the other data are represented as mean ± SEM (error bars).

neurological adverse effects in patients treated with ETC complex I inhibitor IACS-010759[49]. So, it needs further investigations to assess the safety of chemo-activation of *Hs*ClpP for indirect OXPHOS inhibition as a LUSC therapy.

ONC201, a compound undergoing phase III anticancer clinical trials, not only targets mitochondrial ClpP but also induces TRAIL and antagonizes the dopamine receptors D2/D3[50]. Thus, ONC201 is more likely to function as a polypharmacology molecule to exhibit anticancer outcomes, while the exact contribution of targeting *Hs*ClpP is unclear. Though D9 was reported to function as a selective *Hs*ClpP activator in vitro, others found that D9 has a weak activation effect on *Hs*ClpP in cells. Therefore, species-specific, and highly potent activator for *Hs*ClpP is required for full characterization of the druggability of *Hs*ClpP for anticancer therapy. Our activator ZK53 not only selectively activates *Hs*ClpP other than bacterial ClpPs but also shows significant antiproliferative activity in LUSC cell lines. ZK53 does not have a fused-

ring system (like imipridones ONC201 and ONC212) or macrocyclic peptide scaffold (like ADEP 4), which is distinct from those *Hs*ClpP activators typically sharing rigid scaffolds to achieve strong binding affinity (Supplementary Table 5). The similarities and differences between ZK53 and other activators will facilitate further exploration of the druggability of *Hs*ClpP as an anticancer target.

In general, our characterization of the species-specific activator ZK53 provides a mechanism and strategy for the design of selective *Hs*ClpP activators. In the hydrophobic pocket of *Hs*ClpP, the 3,5-difluorophenyl moiety of ZK53 makes π-stacking interactions with the indole ring of W146. Substitution of W146 in human ClpP with isoleucine in bacterial ClpP weakens the π-π stacking effect and causes the leakage of ZK53 when binding in bacterial ClpP in a similar pose. This was confirmed in our mutagenesis studies. Therefore, a highly rigid chemical scaffold in *Hs*ClpP activators is not required for a strong ligand binding, while the activator bearing a simple scaffold also can

achieve selective activation on certain ClpP. It was reported that D190A mutation could be a mechanism of resistance to ClpP activator ONC201[8]. The dependency of ZK53 binding and activation on a single residue (W146) may be a factor in development of drug resistance.

Furthermore, this developmental roadmap, from the discovery of natural product ADEPs to the synthetic imipridones and finally to ZK53 with a more simplified scaffold, represents the molecular evolution of HsClpP activators[51]. Our development of the selective SaClpP activators ZG197 showed that W146 and the joint action with the extra C-terminal motif in HsClpP function as a discriminator for ZG197 enantiomers' selective binding to SaClpP rather than HsClpP, while this work demonstrated that W146 is required for selective ZK53 binding to HsClpP through the π-π stacking effect. These structural features can be used to design different ClpP activators with high species-specificity.

Interestingly, HsClpP activators exhibit antitumor effects but trigger distinct signaling pathways in different cancer cells. Both ONC201 and ZG111 cause significant cell apoptosis. ONC201 induces an atypical integrated stress response in AML cells[8], while ZG111 activates the JNK/c-Jun pathway together with endoplasmic reticulum stress response in PDAC cells[11]. Our selective HsClpP activator ZK53 primarily activates the ATM-mediated DDR signaling pathway and predominantly causes cell cycle arrest in LUSC cells, while the precise mechanism remains unknown. The HsClpP activator ADEP-28 also induces cytotoxic effects by activating intrinsic, caspase-dependent apoptosis in HEK293 T-REx cells[6]. These differences in cell fate determination and regulatory pathways triggered by different HsClpP activators could be attributed to genetic backgrounds in different cancer cell lines and the likelihood of off-target effects and physicochemical properties of these HsClpP activators, which requires further investigation.

We are aware that further illustrations of the mechanism of ZK53-triggered signal transduction between the mitochondria and the nucleus are warranted. Moreover, differences in HsClpP protein abundance do not seem to be simply associated with the inhibitory effects of ZK53 on LUSC cells proliferation. The efficacy of HsClpP activators could differ with various genetic backgrounds of LUSC cells while discovering biomarkers remains challenging. Although no obvious toxicity was observed in mouse models nor ZK53 inhibited gut microbes in vitro, our study does not present adequate information on therapeutic window of ZK53 nor the effects of ZK53 in more non-malignant cell lines at therapeutic dosages, which needs further investigations. In addition, it remains unclear about the tumor biology as well as underlying mechanisms for this metabolic vulnerability for LUSC cancers as the best candidate for certain ClpP activator, such as ZK53. Sex has not been confirmed as a critical factor in this study. We are aware that there may be slight variations in tumor progression between males and females, but for the sake of consistency in the experiments, the same sex was selected for the CDX and KL mouse models. Though no significant difference in expression level of *CLPP* is observed between male and female patients according to the UALCAN dataset, further investigations are required to determine gender differences.

In conclusion, we report on a selective and potent HsClpP activator, ZK53, that shows anticancer effects in LUSC cells, xenograft and autochthonous mouse models. Our study indicates a promising strategy for developing selective HsClpP activators through the evolutionary simplification. The use of the species-specific HsClpP activator will facilitate further investigation into the druggability of HsClpP for targeted cancer therapy.

## Methods

### Ethical statement
The protocols of the xenograft experiments were reviewed and approved by the Institutional Animal Care and Use Committee (IACUC) of the Shanghai Institute of Materia Medica. Experiments were performed following the relevant ethical guidelines and regulations. The laboratory animal usage license (2021-10-YCG08) is certified by IACUC of Shanghai Institute of Materia Medica. For the animal used in the genetically engineered mouse model, all animal procedures were performed under the ethical guidelines of the Center for Excellence in Molecular Cell Science, Chinese Academy of Sciences.

### Animals
Six-week-old female Balb/c mice were purchased from Shanghai Jihui Laboratory Animal Care Co., Ltd. The mice were maintained on an *ad libitum* diet in a specific pathogen-free facility at the Shanghai Institute of Materia Medica, Chinese Academy of Sciences. The $Kras^{LSL\text{-}G12D/+}$;$Lkb1^{fl/fl}$ mice at 6–8 weeks of age were used in the genetically engineered mouse model. The mice were maintained on an *ad libitum* diet in a specific pathogen-free facility at the Center for Excellence in Molecular Cell Science, Chinese Academy of Sciences. Housing conditions included a dark/light cycle of 12 h, an ambient temperature of 20–26 °C, and humidity of 40–60%. Sex has not been confirmed as a critical factor in this study. Although there may be slight variations in tumor progression between males and females, for the sake of consistency in the experiments, the same sex was selected for the CDX and KL mouse models.

### Antibodies
Antibodies of HsClpP (1:2,000, Clo#EPR7133, Cat#ab124822, Lot#GR3210822-7, Abcam), NDUFB8 (1:3,000, Clo#EPR15961, Cat#ab192878, Lot#GR243097-1, Abcam), NDUFA12 (1:2,000, Clo#EPR15867-28, Cat#ab192617, Lot#GR3209820-3, Abcam), SDHA (1:3,000, Clo#5E10G12, Cat#66588-1-Ig, Lot#10006838, Proteintech), SDHB (1:10,000, Clo#EPR10880, Cat#ab175225, Lot#GR3380267-7, Abcam), UQCRC2 (1:2,000, Clo#EPR13051, Cat#ab203832, Lot#GR247396-11, Abcam), COX IV (1:1,000, Cat#ab153709, Lot#GR291729-10, Abcam), p-Rb S795 (1:1,000, Cat#AP0088, Lot#2100940201, Abclonal), p-Rb T826 (1:1,000, Clo#EPR5351, Cat#ab133446, Lot#GR96233-11, Abcam), p-Rb S807/811 (1:1,000, Clo#D20B12, Cat#8516T, Lot#9, CST), Cyclin D1 (1:10000, Clo#EPR2241, Cat#ab134175, Lot#GR3212345-11, Abcam), Cyclin E2 (1:1,000, Clo#ARC1515, Cat#A9305, Lot#4000001515, Abclonal), CDK2 (1:1,000, Cat#A0294, Lot#0600070401, Abclonal), PCNA (1:1,000, Clo# ARC51325, Cat#A12427, Lot#4000002488, Abclonal), γ-H2AX (1:1,000, Clo#20E3, Cat#9718S, Lot#13, CST), p-ATM S1981 (1:1,000, Clo#EP1890Y, Cat#ab81292, Lot#GR3285525-7, Abcam), p-CHK2 T68 (1:1,000, Clo#C13C1, Cat#2197T, Lot#12, CST), p-CHK1 S345 (1:1,000, Clo#133D3, Cat#2348T, Lot#18, CST) CDC25A (1:500, Clo#P30304, Cat#sc-7389, Lot#D2821, Santa Cruz), p-ATR T1989 (1:1,000, Cat#GTX128145, Lot#44384, GeneTex), Ki67 (1:200, Clo#SP6, Cat#ab16667, Lot#GR10004156, Abcam), GAPDH (1:5,000, Clo#1E6D9, Cat#60004-1-Ig, Lot#10025237, Proteintech), β-actin (1:5,000, Clo#2D4H5, Cat#66009-1-Ig, Lot#10004156, Proteintech), SOX2 (1:1,000, Clo#EPR3131, Cat#ab92494, Lot#GR3285529-6, Abcam), Cytokeratin 5 (KRT5) (1:1,000, Clo#XS20200904015, Cat#BS1208, Lot#XCJ36131, Bioworld), p40 (1:200, Clo#ZR8, Cat#RMA-0815, Lot#2108120815C5, MXB Biotechnologies), TTF1 (1:500, Clo#EPR8190-6, Cat#ab133638, Lot#GR3431564-1, Abcam), HRP-conjugated goat anti-rabbit IgG (1:10,000, Cat#CW0103, Cwbio), and HRP-conjugated goat anti-mouse IgG (1:10,000, Cat#CW0102, Cwbio) were commercially purchased. Antibodies of SaClpP (1:5,000, Cat#C11185) and SaGAPDH (1:5,000, Cat#C1399) were generated by Shanghai Immune Biotech Co., Ltd. using the purified proteins as the antigen and validated by ELISA experiments.

### Bacterial strains and mediums
The bacterial strains used in this study are displayed in Supplementary Table 6. Mediums of Tryptic Soy Broth (TSB, OXOID) were used for the cultivation of *S. aureus*. Luria Bertani (LB) in broth or LB agar mediums were used for *E. coli*. *S. aureus* and *E. coli* were routinely cultured at

37 °C. *Limosilactobacillus reuteri* BNCC186563 (LR), *Lactobacillus rhamnosus GG* ATCC53103 (LGG) and *Bifidobacterium longum subsp. Infantis* BNCC185971 (BL) were routinely cultured under anaerobic conditions at 37 °C using an anaerobic cabinet (Whitley A25 Workstation; Don Whitley, 80% $N_2$, 10% $H_2$, 10% $CO_2$) in de Man, Rogosa and Sharpe Broth (MRS medium).

## Cell lines and cultures
The H1703 (male patient-derived, Cat#CRL-5889) and H520 (male patient-derived, Cat#HTB-182) cell lines were obtained from American Type Culture Collection and cultured in PRMI1640 (Corning) supplemented with 10% fetal bovine serum (FBS) (Gibco) and 1% penicillin-streptomycin (Corning); the H226 (male patient-derived, Cat#TCHu235), SK-MES-1 (male patient-derived, Cat#SCSP-5010), MRC-5 (male human-derived, Cat#GNHu41), and HEK293T/17 (Cat#GNHu44) cell lines were obtained from the cell bank of the Chinese Academy of Sciences (Shanghai, China). H226 cells was cultured in RPMI1640 supplemented with 10% FBS, 1% penicillin-streptomycin, 1 mM sodium pyruvate, 1.5 g/L sodium bicarbonate, and 2.5 g/L glucose. SK-MES-1 was cultured in Minimum Essential Medium (Corning) supplemented with 10% FBS, 1% penicillin-streptomycin, 1 mM sodium pyruvate, and 1.5 g/L sodium bicarbonate. HEK293T/17 and MRC-5 cells were cultured in Dulbecco's modified Eagle's medium (Corning) supplemented with 10% FBS, 1% penicillin-streptomycin, and 1.5 g/L sodium bicarbonate. All cells were cultured at 37 °C in a 5% $CO_2$-containing incubator and regularly tested for mycoplasma contamination.

## Chemicals
ZK11, ZK24, and ZK53 were synthesized in the lab and the synthetic methods with the characterizations are provided in Supplementary methods and Supplementary Figs. 7–18. ADEP 4 was purchased from ChemPartner (Shanghai, China). BX471, KU-55933, and AZD1390 were purchased from MedChemExpress and used as received.

## Protein purification and crystallization
The *N*-terminus 56 amino acids truncated wild-type and mutant (W146I) *Hs*ClpPΔ56 without mitochondrial targeting amino acid sequences, *Sa*ClpP (wild-type and I91W), and *Ec*ClpP were expressed and purified as described previously[11,26]. *Lr*ClpP is expressed and purified similarly as *Sa*ClpP. For crystallization, 10 mg/mL purified *Hs*ClpPΔ56 was incubated with ZK53 at a molar ratio of 1:5 (*Hs*ClpPΔ56:ZK53) for 30 min on ice. Then 2 µL of the sample was mixed with 1 µL of crystallization solution (200 mM magnesium acetate tetrahydrate, 20% w/v polyethylene glycol 3350, pH 7.9) using the sitting drop vapor diffusion method. The crystals appeared after 3-4 weeks and were collected and diffracted on the BL02U1 beamline at the Shanghai Synchrotron Radiation Facility. The diffraction data of the ZK53/*Hs*ClpP complex were processed using Aquarium, an automatic data-processing and experiment information management system for biological macromolecular crystallography beamlines[52]. The structure was solved by the molecular replacement method using the Phaser program embedded in the CCP4i suite with the entire heptamer in *Hs*ClpP structure (PDB: 1TG6) as the search model[53,54]. The structure was refined using the refmac5 program of the CCP4i suite[55,56]. The local NCS restraints were automatically applied to the fourteen protomers within the model. The 2fo-fc and fo-fc electron density maps were regularly calculated and used for building the ligands and solvent molecules using COOT[57].

## ClpP enzymatic assays
For the PAGE-based *Hs*ClpP protease activity assay, 0.2 mg/mL *Hs*ClpP, 0.6 mg/mL α-casein (Sigma, C0528), and compounds at indicated concentrations were mixed in the assay buffer (25 mM HEPES-KOH, pH 7.6, 5 mM $MgCl_2$, 5 mM KCl, 0.03% Tween-20, 10% glycerol, and 2 mM DTT). The reaction was incubated at 37 °C for 2 h. The samples were then analyzed by SDS-PAGE and stained with Coomassie brilliant blue. For the $EC_{50}$ calculation, the band intensity was quantified using ImageJ, and $EC_{50}$ was calculated in GraphPad Prism software 8 using the log(agonist) vs. response-variable slope model. As for assaying *Sa*ClpP, *Ec*ClpP, and *Lr*ClpP activity, 0.2 mg/mL *Sa*ClpP, *Ec*ClpP, and *Lr*ClpP each were mixed with compounds in the PD buffer (25 mM HEPES-KOH, pH 7.6, 20 mM $MgCl_2$, 100 mM KCl, 1 mM EDTA, 10% glycerol, and 2 mM DTT), and the following procedures were performed similarly for assaying *Hs*ClpP activity.

The assay buffer used for the FI-based protease activity assay was the same as that used in the PAGE-based protease activity assays for *Hs*ClpP. Compounds with the indicated concentration were incubated with 0.7 µM *Hs*ClpP for 15 min, and then 5 µM FITC-casein (Sigma) was added. After gentle mixing, the fluorescence was quickly monitored for 30 min at 37 °C in a 96-well black flat-bottom plate (Corning) at an interval of 30 s. The initial velocity was defined as the reaction rate of the initial 15 min. The highest velocity was calculated as 100%, and the $EC_{50}$ of protease activity was calculated by GraphPad Prism software 8 using log(agonist) vs. normalized response-variable slope model. As for *Sa*ClpP, 1 µM *Sa*ClpP, 2.5 µM FITC-casein, and PD buffer were used, and other procedures were the same as the *Hs*ClpP assay.

## Differential Scanning Fluorimetry (DSF) assay
The DSF assay was performed as previously described[11]. Briefly, 1 µM *Hs*ClpP, 5 × SYPRO™ Orange (Invitrogen S11368), and compounds at indicated concentration were added to the buffer (20 mM Tris-HCl, pH 7.5, 0.5 M NaCl, 5% glycerol) to a volume of 20 µL. Then the DSF assay was conducted with three replicates on CFX96 Touch Real-Time PCR Detection System (BioRad) with the temperature rising from 25 to 95 °C at a 1% ramp rate. The melting temperature (Tm) was calculated by Bio-Rad CFX Manager 3.1. For the DSF assays on *Sa*ClpP, 2 µM *Sa*ClpP was added to the buffer (50 mM Tris-HCl, pH 8.0, 0.1 M KCl). The other setting is the same as that of *Hs*ClpP.

## Nano Differential Scanning Fluorimetry (NanoDSF) assay
Compound was added to a solution of 1 mg/mL of WT and Y118A *Hs*ClpP proteins in buffer (20 mM Tris-HCl, pH 7.5, 0.5 M NaCl, 5% glycerol). After incubation for 10 min at room temperature, approximately 10 µL of the mixture was loaded to the capillary. The melting curve were monitored by the instrument (NanoTemper, Prometheus NT.48) with the temperature increasing from 25 to 90 °C at a rate of 1.5 °C/min. The first derivative at 330 nm and inflection point were calculated by the software (PR. ThermControl, v2.1.6).

## Growth curve of gut microbes
In brief, LR, LGG and BL were cultured with MRS medium at 37 °C in anaerobic cabinet for overnight. Then, bacteria were collected by centrifugation and washed once by PBS buffer. The bacteria were resuspended and diluted to $OD_{600}$ around 0.2 using MRS medium. Finally, 50 µL bacteria and 50 µL ClpP activators or tetracycline at indicated concentrations (diluted in MRC medium with final DMSO-concentration of 1%) were added in 96-well plates in triplicates and incubated in anaerobic cabinet at 37 °C. After 24 h culture, the turbidity at $OD_{600}$ was measured and recorded by microplates reader (BioTek, USA)

## Lentivirus production and infection
A total of 13.5 µg transgenes (pLKO.1 vector was used for *CLPP* knockdown, pCDH and pLVX-TetOne vectors were used for *CLPP* overexpression), 3.75 µg pMD2.G, 2.25 µg pMDLg/pRRE, and 5.25 µg pRSV-Rev were co-transfected into HEK293T/17 cells in 100-mm dish with Lipofectamine 2000 (Invitrogen, 11668500) according to the manufacturer's instructions. After 48 h and 72 h of transfection, the lentiviruses were harvested, filtered through a 0.45 µm membrane to remove cell debris, and stored at −80 °C. The LUSC cells were infected with indicated lentiviruses for 48 h with 6 µg/mL polybrene

(Genomeditech, GM-040901A). After infection, 1 µg/mL puromycin (Meilune, MB2005) was added to select the positive cells for at least 7 days. The shRNA targetingClpP coding sequence are as follows: shNS: 5′-TGGTTTACATGTTGTGTGA-3shCLPP-1: 5′-GCCCATCCACATGTACAT CAA-3′shCLPP-2: 5′- CACGATGCAGTACATCCTCAA -3′shCLPP-3: 5′-GTT TGGCATCTTAGACAAGGT-3′.

## Immunoblotting
Immunoblotting was performed according to reported methods[58]. The cells were lysed on ice in RIPA Lysis buffer (Beyotime, P0013C) supplemented with protease and phosphatase cocktail inhibitors (Sangon Biotech) for protein extraction. Lysates were centrifuged at $12,000 \times g$ for 20 min at 4 °C, and the pelleted cell debris was removed. Total protein concentration was determined using BCA Protein Assay Kit (Beyotime, P0009), and equal amounts of protein lysates were loaded on SDS-PAGE and transferred onto nitrocellulose membranes (Millipore). After blocking with 5% non-fat milk, the membranes were incubated with indicated primary antibodies at 4 °C overnight and HRP-conjugated secondary antibodies (CWBIO) for 1 h at room temperature. The Chemistar Western Blotting Substrate (Tanon) and ECL detection system (GE Healthcare) were used for visualization.

## Reverse transcription and quantitative real-time PCR (qPCR)
Samples were harvested in TRIzol reagent (Invitrogen, 15596026), and the total RNA was isolated according to the manufacturer's instructions. RNA was reverse-transcribed into cDNA using The HiScript III RT SuperMix for qPCR Kit (Vazyme, R233-01). Then the qPCR was performed using the ChamQ Universal SYBR qPCR Master Mix (Vazyme, Q711-02) and was run on a CFX96 Touch Real-Time PCR Detection System (BioRad). Ct values were normalized against the housekeeping gene *GAPDH*. The primer sequences in this study are provided in the Supplementary Table 7.

## Clonogenic assay
Approximately 1000–1500 LUSC cells or stably transfected cells were seeded into 12-well plates in triplicate and allowed to adhere for 24 h. Then indicated compounds (or doxycycline) were added, and the cells were continued to grow for 10–14 days. Afterward, the cells were washed with PBS, fixed with 10% acetic acid and 10% methanol for 15 min, and stained with 1% crystal violet in 25% methanol for 1 h. After washing out the excessive crystal violet, the plates were photographed. For quantification, the stain was extracted with 10% acetic acid and 10% methanol, and the absorbance at 590 nm was measured.

## Cell proliferation assays
Approximately 1500–5000 LUSC cells were seeded into 96-well plates and allowed to adhere for 24 h. Then the cells were treated with indicated compounds for 72 h. The MTT solution (Meilunbio, MA0198) was added to the cells and incubated at 37 °C in a 5% CO₂-containing incubator for 4–6 h. Afterward, the supernatant was removed, and the formazan product was dissolved with DMSO. The absorbance of the formazan solution was read at 490 nm. As for the GI$_{50}$ determination, additional cell viability was obtained at the time point of compound treatment ($t = 0$). For the proliferation curve of Dox-induced *Hs*ClpP overexpression, the cells were seeded into 96-well plates with (or without) 1 µg/mL Dox. Then the cell viability was determined at indicated time points. For the proliferation curve of compound treatment, the cells were seeded into 96-well plates, and 24 h later, the compound was added. Then the cell viability was determined at indicated time points.

## EdU proliferation assay
The LUSC cells were seeded into 12-well dishes and allowed to adhere for 24 h before adding ZK53. After 48 h of treatment, the samples were prepared for EdU proliferation assay using the Meilun EdU Cell Proliferation Kit with Alexa Fluor 488 (Meilunbio, MA0424-1) according to the manufacturer's instructions. The samples were then analyzed through an Olympus IX73 fluorescence microscope.

## Cell cycle analysis
The LUSC cells were seeded into 60-mm dishes and allowed to adhere for 24 h before adding ZK53. After 48 h, the samples were prepared for cell cycle analysis using the Cell Cycle and Apoptosis Analysis Kit (Meilunbio, MA0334) according to the manufacturer's instructions. The samples were then analyzed by a BD FACSCalibur flow cytometer, and the cell cycle phase distribution was analyzed by ModFit LT software.

## Apoptosis analysis
The LUSC cells were seeded into 6-well plates. After adhering for 24 h, 1 µM ZK53 was added, and the cells were treated at different times as indicated. Then the cells were harvested, washed with PBS, and stained with annexin V-FITC using the Annexin V-FITC/PI Apoptosis detection Kit (Meilunbio, MA0220). The samples were analyzed by a BD FACS-Calibur flow cytometer. The proportion of apoptotic cells (annexin V-positive) was analyzed using FlowJo software.

## Extracellular flux analysis
The LUSC cells were seeded into an XFe96 cell culture microplate (Agilent) and allowed to adhere for 24 h. Then the cells were treated with ZK53 at indicated concentrations for 48 h. The Seahorse XF Cell Mito Stress Test Kit (Agilent, 103015-100) was used according to the manufacturer's instructions, and the oxygen consumption rate (OCR) was then monitored with a Seahorse XFe96 Analyzer (Agilent).

## Mitochondrial membrane potential (MMP) measurement
For the MMP measurement, the Enhanced Mitochondrial Membrane Potential Assay Kit with JC-1 (Beyotime, C2003S) was used according to the manufacturer's instructions. Briefly, after treatment with ZK53 for 72 h, the LUSC cells were collected and resuspended in a JC-1 working solution. The cells were incubated for 20 min at 37 °C in the dark, washed three times, and analyzed using the BD FACSCalibur flow cytometer. The relative MMP level was calculated using the red/green fluorescence intensity median and normalized to the DMSO-treated group.

## ATP level determination
The LUSC cells were seeded into 12-well plates. After adherence, ZK53 was added, and the cells were treated for 72 h. The Enhanced ATP Assay Kit (Beyotime, S0027) was used according to the manufacturer's instructions. ATP levels were normalized to the DMSO-treated group.

## Mitochondrial ROS measurement
The LUSC cells were seeded into 6-well plates, and after adherence, the cells were treated with ZK53 at the indicated concentration. After treatment for 48 h, the cells were collected and washed three times with pre-warmed Hanks Balanced Salt Solution (HBSS, Meilunbio, PWL056_A). Then the cells were treated with 5 µM MitoSOX (Invitrogen, M36005) in the dark at 37 °C for 20 min and washed with HBSS. The fluorescence intensity was determined by BD FACSCalibur flow cytometer and analyzed using FlowJo software.

## Mitochondrial DNA (mtDNA) copy number measurement
The mtDNA copy number measurement was conducted as previously reported[59]. The cells treated with ZK53 for 48 h were collected and washed with PBS. Then the total DNA was isolated using the DNeasy Blood & Tissue Kit (Qiagen, 69504). The relative mtDNA copy number was measured by qPCR using the primers for amplifying the *MT-ND1* gene in mtDNA and primers for amplifying the *HGB* gene in the nuclear

genome. qPCR was conducted using 75 µg DNA. Primer sequences are listed in Supplementary Table 7.

## Cellular thermal shift assay (CETSA)

The intact cell CETSA was performed according to the previous study[29]. Briefly, LUSC cells were treated with DMSO or 10 µM ZK53 for 2 h. Then the cells were harvested and aliquoted into 0.2 mL PCR tubes. After denaturing at different temperatures for 3 min and equilibration at room temperature for 5 min, the samples were freeze-thawed three times using liquid nitrogen. Then the soluble fraction was collected by centrifugation at $12,000 \times g$ for 20 min at 4 °C and analyzed by immunoblot analysis. For CETSA of *S. aureus* Newman, overnight cultures of Newman were diluted at 1:100 in TSB and were treated with DMSO or 10 µM indicated compounds for 2 h. Then the cells were harvested and aliquoted into 0.2 mL PCR tubes. After denaturing at different temperatures for 3 min and equilibration at room temperature for 5 min, the samples were lysed with 0.75 U lysostaphin at 37 °C for 30 min, followed by three freeze-thaw cycles with liquid nitrogen. The following procedures are the same as the LUSC cells.

## RNA-sequencing

H1703 cells were treated with 1 µM ZK53 for 48 h before being lysed using the TRIzol reagent (Invitrogen). All samples were frozen at −80 °C before RNA extraction and analysis. The following steps were conducted by Majorbio Bio-Pharm Technology Co., Ltd. (Shanghai, China). Total RNA was extracted, and DNase I (Takara) was used to remove the genomic DNA. Then RNA-seq transcriptome library was prepared using a TruSeq™ RNA sample preparation Kit (Illumina, San Diego, CA) with 1 µg of total RNA. Briefly, mRNA was isolated and fragmented, and double-strand cDNA was synthesized using the SuperScript double-stranded cDNA synthesis kit (Invitrogen, CA) with random hexamer primers (Illumina). Afterward, as recommended by Illumina's library construction protocol, cDNA was end-repaired, phosphorylated, and added 'A' base. After size selection for cDNA target fragments of 300 bp, the paired-end RNA-seq sequencing library was sequenced with the Illumina HiSeq xten/NovaSeq 6000 sequencer (2 × 150 bp read length). For read mapping, the raw reads were trimmed and quality controlled by SeqPrep (https://github.com/jstjohn/SeqPrep) and Sickle (https://github.com/najoshi/sickle). The clean reads were aligned to a reference genome with orientation mode using HISAT2 (http://ccb.jhu.edu/software/hisat2/index.shtmL)[60] software, and the mapped reads were assembled by StringTie (https://ccb.jhu.edu/software/stringtie/index.shtml?t=example) in a reference-based approach[61]. The RNA-seq analysis was performed at the online platform Majorbio Cloud Platform (www.majorbio.com).

## Immunofluorescence

After treatment with 1 µM ZK53 for 48 h, the LUSC cells were fixed with 4% paraformaldehyde for 15 min and washed three times with PBS. After permeabilization with 0.1% Triton X-100 (Sigma-Aldrich, 93443) for 15 min and blocking with 10% goat serum for 1 h, the cells were incubated with the γ-H2AX antibody at 4 °C overnight. Then the cells were washed three times, and the FITC-conjugated secondary antibody (Invitrogen, A11034) was applied for 2 h. Finally, the cells were washed, and the nucleic acid was counterstained by 1 µg/mL DAPI (Invitrogen, 62248). The Leica TCS SPS CFSMP confocal microscopy was used for imaging.

## Alkaline comet assay

The alkaline comet assay was conducted for DNA damage detection using the OxiSelect™ Comet Assay Kit (Cell Biolabs, STA-351-5) according to the manufacturer's instructions. Briefly, H1703 cells were seeded into 6-well plates and treated with 2.5 µM ZK53 for 72 h. Then the cells were collected, mixed with agarose, lysed with lysis buffer, and immersed in the alkaline solution. Then the slides underwent

alkaline electrophoresis for 20 min, and the DNA was stained using the vista green DNA dye and visualized with the Olympus IX73 fluorescence microscope.

## Xenograft mice model

To assess Y118A ClpP overexpression on tumor growth in the animal model, H1703 cells transfected with Dox-inducible Y118A ClpP ($7 \times 10^6$ per mouse) were mixed with 50% Matrigel (Corning, 354234), and 100 µL of the mixture was subcutaneously inoculated into the flank of 4–6-week-old female Balb/c nude mice. After the tumor volume reached approximately 80 mm³, the mice were randomly placed into two groups and treated with or without 2 mg/mL Dox in drinking water. The tumor volume (mm³) was calculated using the formula $V = \text{length (mm)} \times \text{width (mm)}^2/2$. The p values were determined by a two-tailed Student's t test.

To assess the effects of ZK53 in vivo, H1703 cells ($7 \times 10^6$ per mouse) were mixed with 50% Matrigel (corning), and 100 µL of the mixture was subcutaneously inoculated into the flank of 4–6-week-old female Balb/c nude mice. After the tumor volume reached approximately 80 mm³, the mice were randomly grouped, and 80 mg/kg ZK53 (dissolved in 40% 2-hydroxypropyl-β-cyclodextrin) was administered intraperitoneally twice a day, five days a week. The tumor volume (mm³) was calculated using the formula $V = \text{length (mm)} \times \text{width (mm)}^2/2$. The p values were determined by a two-tailed Student's t test.

The maximal tumor size permitted by the Institutional Animal Care and Use Committee (IACUC) of the Shanghai Institute of Materia Medica is 2000 mm³. The maximal tumor size in this paper was not exceeded to this limit.

## H&E staining

The tumors and organs were fixed in 4% formaldehyde and processed for paraffin embedding. Four-micrometer sections were cut and mounted on glass slides. Then the slides were counterstained with hematoxylin and eosin.

## Complete blood content analysis and plasma biochemical analysis

The PB samples were collected for the complete blood content analysis and plasma biochemical analysis (Shanghai Meixuan Biological Science & Technology Ltd., China).

## Genetically engineered mouse model

The *Kras*[LSL-G12D/+]*;Lkb1*[fl/fl] mice at 6–8 weeks of age were treated with Ad-Cre ($2 \times 10^6$ p.f.u) via nasal inhalation as previously described[46]. Treatment of ZK53 was conducted twice a day for 4 weeks through intraperitoneal injection. Mice were sacrificed for histopathological examination, and tumor burden was analyzed using ImageJ software as previously described[62].

## Statistics and reproducibility

Data are presented as mean ± SD or mean ± SEM as indicated. All statistical analyses were performed using GraphPad Prism 8.3 software. P values are indicated in the figures, and $P < 0.05$ was considered significant. The number of independent experiments/mice/samples is mentioned in the figure legends. All attempts at replication were successful. All blots and gels were performed in triplicate, and a single experimental image is shown. Xenograft and genetically engineered mouse experiments, H&E staining, and immunohistochemistry (IHC) were performed once. Three representative images of H&E staining and IHC were taken in each sample, and one is shown. All images recorded on H&E staining and IHC indicated a similar trend.

## Reporting summary

Further information on research design is available in the Nature Portfolio Reporting Summary linked to this article.

## Data availability

The atomic coordinates and structure factors data generated in this study have been deposited in the Protein Data Bank (PDB, www.pdb. org) under accession code 8HGK for ZK53/*Hs*ClpP. Other X-ray structural data used in this study are available in the PDB database under accession code 1TG6. The amino acid sequence can be found at the National Center for Biotechnology Information (NCBI, https://www. ncbi.nlm.nih.gov/) with the accession number NP_006003 for *Hs*ClpP; KFL07692 for *Sa*ClpP. The processed The Cancer Genome Atlas Program (TCGA) data are available under restricted access for copyright, access can be obtained by UALCAN (https://ualcan.path.uab.edu/ index.html) created by the university of Alabama at Birmingham. The RNA-seq data generated in this study have been deposited in NCBI SRA dataset under accession code PRJNA902171. Source data are provided with this paper.

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

## Acknowledgements

We thank the staff from BL02U1 beamline at Shanghai Synchrotron Radiation Facility for data collection support. This work was financially supported by the National Key Research and Development Program of China (2022YFC2804100 to C.-G.Y., 2022YFA1103900 to H.J.), the National Natural Science Foundation of China (22037007 and 21725801 to C.-G.Y., 22107109 to T.Z., 82341002, 32293192 and 82030083 to H.J.), and the Youth Innovation Promotion Association of CAS (2023297 to T.Z.).

## Author contributions

C.-G.Y. conceived the project and supervised the research with help from H.J; L.-L.Z. conducted research with the help from T.Z., Y.X., C.Y., Y.P., P.W., T.Y. M.L., H.Z., K.D., and J.G.; T.Z. synthesized activators; Y.X. performed the in vivo experiment in autochthonous mouse models; H.Z., K.D., J.G., H.J., and C.-G.Y. contributed agents and data analysis; L.-L.Z. and C.-G.Y. wrote the manuscript with help from T.Z. and Y.X. All authors discussed the results and commented on the manuscript.

## Competing interests

L.-L.Z., T.Z. and C.-G.Y. are named inventors of pending patent application (2023106791890, to the Chinese Patent Office) related to the work described. Patent applicant is Shanghai Institute of Materia Medica, Chinese Academy of Sciences. The structures of ZK11 and ZK53 are covered in the patent application. The remaining authors declare no competing interests.
