## [Peer Review File · Nature Communications]

REVIEWER COMMENTS

Reviewer #1 (Remarks to the Author):

Zhou et al. describe in this manuscript the anti-tumor activity and mechanism of action of ZK53 against lung squamous cell carcinoma. Previously, imipridone compounds and D9 have been demonstrated to be the HClpP activators, however imipridone compounds showed no species selectivity, and the HClpP selective activator D9 demonstrated no anti-tumor effects in cancer cells in this report and in hands of this reviewer. The authors have synthesized a new compound named ZK53 that has both antitumor activities and species selectivity. According to their results, ZK53 displays high activity for its target, the human mitochondrial protease ClpP, and shows in vitro and in vivo antitumor activity. Overall, this manuscript is well organized and the finding is important for the HClpP drug discover field. I suggest to accept this manuscript for publication in nature communications after addressing the following issues.

1. In CETSA assay, cell proteins were collected after 2 h of drug treatment of the cells, but in the literature it is more common to extract the proteins first and then co-incubate the drug, which seems to be more applicable because the membrane permeability of the drug does not need to be taken into account.

2. In Fig 3, the cell cycle assay does not seem to fit well and needs to be repeated to validate the results.

3. The concentrations of ZK53 were not consistently in vitro cellular assays, such as OC, ATP and ROS assays, why?

4. BX471 has been demonstrated as a selective antagonist of the CC chemokine receptor-1. This effect of ZK53 on CC chemokine receptor-1 should be explored.

5. It has been reported in the literature that Y118A ClpP increases the affinity for agonists. According to the data in this manuscript, Y118A inhibited tumor growth. What is the underlying mechanism?

Reviewer #3 (Remarks to the Author):

In this work, the authors report the development of ZK53, a specific activator of HsClpP, where the crystal structure ZK53/ClpP showed the importance of W146 residue of ClpP in the selective binding with the difluorobenzyl motif of the activator. The mode of action of ZK53 was further assessed by showing the inhibition of LUSC cell proliferation, by targeting the ETC subunits of mitochondrial proteins. Additional relevant data supported the effect of ZK53 on the essential regulatory pathways for G1/S transition, and its activation of the ATM-mediated DNA damage. Finally ZK53 was shown to exhibit anticancer activity in in vivo mice model. Collectively, all the data reported provided enough evidence regarding the druggability of HsClpP as an anticancer target in LUSC therapy. I recommend this paper to be accepted. I do only suggest in the discussion part, to provide clearer points of comparison between ZK53 and other ClpP activators, such as ONC201, ADEP 28, for example in a table form, with the precise reported mechanism of action, selectivity, whether flexible or rigid scaffold,.. just to emphasize the similarities and differences between ZK53 and the other activators

Reviewer #4 (Remarks to the Author):

The study by Zhou et. al. entitled by "Selective activator of human ClpP triggers cell cycle arrest to inhibit lung squamous cell carcinoma" describes a newly developed activator of caseinolytic proteinase P (ClpP), ZK53 that exhibits surprisingly strong anti-cancer effects in a human lung squamous xenograft model. Given its well-established role for mitochondrial proteome homeostasis, ZK53 appears to promote the degradation of components of mitochondrial electron transfer chain resulting in disruption of oxidative respiration and ATP generation in squamous cancer cells. The authors further demonstrate that ZK53 suppresses E2F target network and induces cell-cycle arrest although the molecular mechanism underlying the connection between ZK53-mediated mitochondrial alteration and E2F signaling is unclear.

Considering therapeutic potential of anti-OXPPOS for cancer therapeutics, this study provides a

clinically relevant proof-of-concept of targeting mitochondrial bioenergetic functions for cancer therapy. However, a recent study that reported serious neurological adverse effects in patients treated with complex I inhibitor (IACS-010759, Yap et. al. Nat Med, 29, 115-126, 2023) significantly weakens the enthusiasm of this study. Although the authors argue that no obvious toxicities have been noted in histology and blood biochemical analysis, this study does not present information on therapeutic window of ZK53 nor the effects of ZK53 in non-malignant cells (e.g. human bronchial epithelial cells). It would be critical to show ZK53 has significantly less or no mitochondrial OXPHOS inhibition as well as cytotoxic effects on normal tissues.

Another critical weakness of his study is to depend entirely on cell line system, which significantly weakens the clinical relevance. It would be extremely interest to validate its anti-cancer effects on autochthonous mouse models such as KrasG12D; LKB1-null or LKB1-null; PTEN-null mice. These two systems have been well-employed by a number of studies and reported to develop lung squamous carcinoma with other types of lung cancers.

It is unclear why lung squamous cancer has been selected for this study. Is lung squamous cancer most sensitive to ZK53? If so, what would be underlying mechanisms for this metabolic vulnerability? The authors need to provide biological as well as clinical rationale for lung squamous cancers as the best candidate for ZK53.

Reviewer #2 (Remarks to the Author):

Selective activator of human ClpP triggers cell cycle arrest to inhibit lung squamous cell carcinoma

Zhou et al

Following on from the recent successes (clinical trials) of imipridone action on ClpP in selective cancer therapies, and the group's own recent work on Clp activators reported in Cell Chemical Biology last year, the manuscript by Zhou et al presents a novel ClpP-activating ligand (ZK53), designed using an unrelated chemical scaffold. The druglike molecule binds to a specific pocket on ClpP, increases its thermostability and enzymatic activity and shows anti-proliferation activity.

The basis of the study is not new, as the effects of targeting ClpP and ETC in cancer are documented, but the experimental work is methodologically sound and of good quality. Standard protocols are employed to analyse ClpP activity, mitochondrial function, etc. Gene expression and gene set variation analyses were used to identify the wider reactome, showing gene sets most affected by ZK53 are e2f transcription factors and genes involved in cell cycle progression. Using a specific inhibitor, they implicated ZK53/ClpP in ATM-specific DNA damage response pathways. Analyses showing the effects of ZK53 on mitochondrial function are expertly performed, with appropriate statistical analysis.

Weaknesses:

The authors need to make a cogent case that ZK53 improves on, or provides a viable alternative to, previous ClpP activators in order to demonstrate the significance of their work to the field. Without this, its noteworthiness is unclear. Their key points are that ZK53 is specific for human ClpP, has a flexible chemical scaffold and that it interferes with key mitochondrial functions (e.g. OXPHOS). Please see below for more specific comments on these. Given that ZK53 blocks ClpP and has anti-proliferative activity, an end in itself, the emphasis on whether ZK53 inhibits OXPHOS needs to be clearly set out.

The experiments here specifically address effects of ZK53 on cell lines and a mouse model. A weakness is that the experimental data is discussed more-or-less in isolation, given that effects of high ClpP activity on mitochondria and the ETC are already known. The authors are also rather too selective about what they test for. A comparison of the effects of ZK53 to ClpP activators (the imipridone ONC201 in particular) would have led to more defined, informative, outcomes, e.g. a proteomic comparison with ONC201 to indicate if these quite different druglike molecules disrupt similar cell pathways (if they don't concur, then there is an immediate handle on the differences). If ZK53 is acting solely on ClpP, one might also expect similar outcomes to the constitutively active mutant Y118A (ref 8: Ishizawa, J., *et al.* Mitochondrial ClpP-Mediated Proteolysis Induces Selective Cancer Cell Lethality. *Cancer Cell* **35**, 721-737 2019).

Discussion. The first paragraph, final sentence is a conflation of different streams of data and not proven. While they may have 'demonstrated that ZK53-activated *HsClpP* degrades the ETC subunits and impairs the OXPHOS in LUSC cells', to say "our study provides a proof of concept that using chemo-activation of *HsClpP* [for attenuation of mitochondrial energy metabolism] can be used as a LUSC therapy" is drawing a long bow. The part in square parentheses should be omitted. The ClpP activation clearly causes many changes and there is some work ahead yet to fully appreciate which are of most relevance to tumours.

Other points

The introduction should include an explanation for why specificity for human over bacterium ClpP is desirable in this anti-cancer drug context. If we were looking in an antibiotic context, specificity for

bacteria over Hs would be a key consideration. Here it isn't. Or isn't explained well. In this context of cancer treatment, specificity of action against other cellular/mitochondrial proteins, and in vivo effects of ClpP upregulation on tumour cells vs normal cells, would be far more useful criteria (for avoiding off-target effects).

In the statement (Page 3 para 3) "In this study, we develop a highly selective and potent *HsClpP* activator ZK53", it's important to qualify selectivity as either protein or organism selectivity. Is selectivity only compared to *S. aureus* and *E. coli*, or to other organisms or proteins?

Could you define the proximity of the active site of the protease to the allosteric drug binding site in Figure 1 or 2 please.

The authors have shown that ZK53 is much better at reducing tumour volume than Y118A. Given that Y118 is a peripheral part of the drug-binding pocket, it would seem relevant to understand whether it still binds the drug and if this is still effective. Inclusion of Y118A data in Fig 2f-g would strengthen the study.

What do the authors mean (specifically) by 'flexibility' of ZK53? Do they mean free rotation about the single C-C bonds? In general, small molecules such as these have limited flexibility, adopting the lowest energy conformation, whether bound to a protein or not. It might be more accurate to say that ZK53 does not have a fused ring system (like imipridones), if that is what is meant.

Following on, Page 14 states "we assume that ZK53 can freely approach the hydrophobic pockets of *HsClpP* and adopts a fixed configuration using an essential π - π stacking effect between the difluorobenzyl moiety and W146 of *HsClpP*". If I am reading it correctly, this assumption is not consistent with knowledge of small molecules (as above).

Page 3. "However, ADEPs also activated *HsClpP* and caused mammalian cell death". This was stated as a defect, which it would be - if they were dealing with antibiotics.

Discuss whether ZK53 also kill mammalian cells at therapeutic dosages? The EC_{50} is close to that of ADEP 4 in Fig 2f.

Have you tested ZF53 on non-xenografted control mice?

Page 6 and Fig. 2. The isoleucine mutant W146I does not bind ZK53 (by thermostability), due to an inability to pi-stack with the drug. This would seem to be of concern in terms of potential drug resistance. How conserved is that residue (W146) across eukaryotic species, and could this be a drawback of this simpler molecular scaffold with only a single critical interaction (π - π)? The Y118A mutant is evidence that the binding pocket can adapt.

Page 9. The choice of proteins for Fig 4 appears limited to ETC members. If the choice is based on published data, a comparison with the literature is warranted. Otherwise, why are proteins from other mitochondrial compartments not also selected for testing? It would appear they have already decided what the answer may be (based on prior studies) and are not testing other proteins. Experimental choices made throughout are selective.

Page 12. "The major organs in the endpoint of treatment were subjected to hematoxylin and eosin (H&E) staining, and no noticeable pathological changes were observed between the vehicle and ZK53-treated mice (Supplementary Fig. 6b)." Is this the case for the lung? The vehicle and ZK53 histology are significantly different.

The heading of Supplementary Fig 6 'ZK53 exhibits antitumor activity in vivo' doesn't reflect the data shown. The caption requires much more detail and should state if these are all xenograft mice.

Crystal structure analysis.

The PDB report supplied is a preliminary one – annotated “Not for Manuscripts”, indicating a full deposition has not yet been made. Importantly, the ligands (ZK53) central to the manuscript are not included. A final deposition report should be supplied.

Given the high apparent symmetry of the complex (two heptamers), was there a reason for individual refinement of the 14 chains throughout (without applying NCS averaging at any stage)? The methods section should detail the protocols used in refinement and specify particulars of the model - such as chain breaks.

For a structure refined to 1.9 Å resolution, the B-factors are a bit higher than would be expected, and I would also have expected more water molecules built in. Could you include examples of the electron density maps in the Supplementary Information please.

The report indicates a 2.4% twinning fraction. Was this treated appropriately?

Response to the reviewers' comments

Title: Selective activator of human ClpP triggers cell cycle arrest to inhibit lung squamous cell carcinoma

Manuscript ID: NCOMMS-22-48226A

SUMMARY: We are grateful to the reviewers for the positive recommendations, and we appreciate the thoughtful and constructive suggestions. We have carefully revised our manuscript and addressed all the concerns. The following is a summary of the major changes or modifications during our revision.

1) We assessed the inhibitory effects of our activators on the viability of non-malignant cells. ZK53 displayed less cytotoxicity on the human lung fibroblast MRC-5 cells than the LUSC tumor cells. While it should be aware that the therapeutic window of ZK53 in LUSC treatment and the effects of ZK53 in non-malignant cells at therapeutic dosages still need more investigations in the future.

2) We evaluated the chemo-activation effects of our activators on the ClpP protein of gut microbes and the inhibitory effects on the isolated gut microbes. ZK53 did not activate the ClpP protein of gut microbes and did not show bactericidal effects on the isolated gut microbes *in vitro*.

3) We have performed additional *in vivo* experiments to evaluate the anticancer efficacy of our activator in the *Kras^{LSL-G12D/+};Lkb1^{fl/fl}* genetically engineered mouse model. ZK53 exerted significant anticancer effects in this *in vivo* model.

The following are our point-by-point responses to the comments/suggestions from the reviewers:

1. Response to the comments of Reviewer 1:

General Comments from Reviewer 1: Zhou et al. describe in this manuscript the antitumor activity and mechanism of action of ZK53 against lung squamous cell carcinoma. Previously, imipridone compounds and D9 have been demonstrated to be

the HClpP activators, however imipridone compounds showed no species selectivity, and the HClpP selective activator D9 demonstrated no antitumor effects in cancer cells in this report and in hands of this reviewer. The authors have synthesized a new compound named ZK53 that has both antitumor activities and species selectivity. According to their results, ZK53 displays high activity for its target, the human mitochondrial protease ClpP, and shows in vitro and in vivo antitumor activity. Overall, this manuscript is well organized and the finding is important for the HClpP drug discover field.

Response: We thank the reviewer for the positive comments. We have carefully revised our manuscript.

Comments 1 from Reviewer 1: In CETSA assay, cell proteins were collected after 2 h of drug treatment of the cells, but in the literature, it is more common to extract the proteins first and then co-incubate the drug, which seems to be more applicable because the membrane permeability of the drug does not need to be taken into account.

Response: Thanks for the suggestions. We performed the CETSA assay using the cell lysate of H1703 cells. As shown in Fig. 4g, ZK53 treatment dramatically increased the thermal stability of the *HsClpP* protein in the cell lysate of H1703 cells.

Comments 2 from Reviewer 1: In Fig 3, the cell cycle assay does not seem to fit well and needs to be repeated to validate the results.

Response: Thanks for showing us this mistake. We repeated the cell cycle assay and fitted the model well. The graphs in Fig. 3h-i have been updated in the revision.

Comments 3 from Reviewer 1: The concentrations of ZK53 were not consistently in vitro cellular assays, such as OCR, ATP and ROS assays, why?

[redacted]

Comments 4 from Reviewer 1: BX471 has been demonstrated as a selective antagonist of the CC chemokine receptor-1. This effect of ZK53 on CC chemokine receptor-1 should be explored.

Response: We agree with the reviewer. We have tried to set up the assays to test whether BX471 and the derivative ZK53 inhibit CC chemokine receptor-1 (CCR1). Due to technical limitations in studying GPCR proteins in our lab, unfortunately, we failed to determine the effect of ZK53 on CCR1. BX471 has been used as a potential drug for the treatment of chronic inflammatory diseases, and BX471 has low cellular toxicity, as reported (*J Biol Chem*, **2000**, 275, 19000-8).

At this moment, we cannot exclude the possibility of ZK53 similarly acting as a CCR1 antagonist. However, we have adequately demonstrated the target engagement of our activator on the antiproliferation by primarily targeting *HsClpP* in LUSC cells. The abundance of the *HsClpP* protein substantially influenced the sensitivity of LUSC tumor cells to our activators. Knockdown of *CLPP* significantly decreased the antiproliferative effects of ZK53 in LUSC cells, while overexpressing *CLPP* increased the inhibitory effects of ZK53. These findings suggest that *HsClpP* is the primary target in LUSC cells. To clarify this concern, we have added a statement in the revision, “Though the inhibition of CCR1 was not associated with anticancer (*J Biol Chem*, **2000**, 275, 19000-8), the *in vitro* inhibitory effect of ZK53 on CCR1 should be explored in future.”.

Comments 5 from Reviewer 1: It has been reported in the literature that Y118A ClpP increases the affinity for agonists. According to the data in this manuscript, Y118A inhibited tumor growth. What is the underlying mechanism?

Response: It has been reported that Y118A ClpP mutant functions as a self-activated protease to unselectively degrade multiple proteins, leading to tumor cell death, which phenocopies the activator-promoted HsClpP (*Cancer Cell*, **2019**, 35, 721-37 e9; *Cell Chem Biol*, **2022**, 29, 1396-408.e8). Under this mechanism, it is unsurprised that in this study we found that Dox-induced Y118A ClpP overexpression exhibits antitumor effects in LUSC cells and xenograft mouse models (Fig. 3a-b and Fig. S3a-b). We also found that the Y118A ClpP mutant is capable of degrading α -casein protein *in vitro* (Fig. S2g), though this activity is moderate compared with ClpP activators. To test whether Y118A ClpP increases the binding affinity for activators, we performed the NanoDSF assay to determine the thermal stabilization effects of ClpP activators on the Y118A ClpP mutant. As Fig. S2i shows, except ADEP 4, most activators displayed comparable thermally stabilized effects on ClpP WT and the Y118A mutant *in vitro*.

2. Response to the comments of Reviewer 2:

General Comments from Reviewer 1: Following on from the recent successes (clinical trials) of imipridone action on ClpP in selective cancer therapies, and the group's own recent work on Clp activators reported in *Cell Chemical Biology* last year, the manuscript by Zhou et al presents a novel ClpP-activating ligand (ZK53), designed using an unrelated chemical scaffold. The druglike molecule binds to a specific pocket on ClpP, increases its thermostability and enzymatic activity and shows anti-proliferation activity.

The basis of the study is not new, as the effects of targeting ClpP and ETC in cancer are documented, but the experimental work is methodologically sound and of good quality. Standard protocols are employed to analyse ClpP activity, mitochondrial function, etc. Gene expression and gene set variation analyses were used to identify the wider reactome, showing gene sets most affected by ZK53 are e2f transcription

factors and genes involved in cell cycle progression. Using a specific inhibitor, they implicated ZK53/ClpP in ATM-specific DNA damage response pathways. Analyses showing the effects of ZK53 on mitochondrial function are expertly performed, with appropriate statistical analysis.

Response: We thank the reviewer for the positive recommendation. We have carefully revised our manuscript and provided a point-by-point response to the specific comments.

Comments 1 from Reviewer 2: The experiments here specifically address effects of ZK53 on cell lines and a mouse model. A weakness is that the experimental data is discussed more-or-less in isolation, given that effects of high ClpP activity on mitochondria and the ETC are already known. The authors are also rather too selective about what they test for. A comparison of the effects of ZK53 to ClpP activators (the imipridone ONC201 in particular) would have led to more defined, informative, outcomes, e.g. a proteomic comparison with ONC201 to indicate if these quite different druglike molecules disrupt similar cell pathways (if they don't concur, then there is an immediate handle on the differences). If ZK53 is acting solely on ClpP, one might also expect similar outcomes to the constitutively active mutant Y118A (ref 8: Ishizawa, J., et al. Mitochondrial ClpP-Mediated Proteolysis Induces Selective Cancer Cell Lethality. *Cancer Cell* **35**, 721-737 2019).

Response: Thanks for the great suggestion. We agree with the reviewer and we have performed additional proteomic comparisons with ONC201 in LUSC cells. As shown in Fig. S5g-h, both ZK53 and ONC201 downregulate Rb phosphorylation and the regulatory proteins required for Rb phosphorylation, activate the DDR pathway, and inhibit the E2F targets transcript level in LUSC cells. We also checked whether ZK53 regulates the reported ONC201-activated pathway, including Akt-Erk, ATF4, and DR5 in LUSC cells (*Science Translational Medicine* **5**, 171ra17 2013; *Science signaling* **9**, ra18 2016). We found that ZK53 indeed similarly regulated these cell pathways as ONC201 did in LUSC cells (Fig. S5i). Collectively, these data indicate that the two quite different druglike molecules, ZK53 and ONC201 disrupt similar cell pathways in

LUSC cells.

In addition, we tested whether the constitutively active Y118A ClpP mutant has similar outcomes with ZK53. The immunoblot results showed that the Y118A ClpP mutant induced similar changes of key proteins as ZK53 (Fig. S5j) in LUSC cells. Since Y118A mutant exhibited a much weaker *in vitro* activity compared with small-molecule activators (please also see the response to comments 5 from Reviewer 1) (Fig. S2g), it is unsurprised that the constitutively active Y118A ClpP is much weaker than the chemo-activation of ClpP for disruption on the cellular proteins. Thus, ZK53 shows similar outcomes to the constitutively active mutant Y118A, which reveals the cellular target engagement of ZK53 in LUSC cells.

Comments 2 from Reviewer 2: Discussion. The first paragraph, final sentence is a conflation of different streams of data and not proven. While they may have 'demonstrated that ZK53-activated HsClpP degrades the ETC subunits and impairs the OXPHOS in LUSC cells', to say "our study provides a proof of concept that using chemo-activation of HsClpP [for attenuation of mitochondrial energy metabolism] can be used as a LUSC therapy" is drawing a long bow. The part in square parentheses should be omitted. The ClpP activation clearly causes many changes and there is some work ahead yet to fully appreciate which are of most relevance to tumours.

Response: We have reformatted the sentence as "Different from the primary approach of direct inhibition on ETC complexes with small-molecule inhibitors, we demonstrated that ZK53-activated HsClpP degrades the ETC subunits and impairs the OXPHOS in LUSC cells."

Comments 3 from Reviewer 2: The introduction should include an explanation for why specificity for human over bacterium ClpP is desirable in this anticancer drug context. If we were looking in an antibiotic context, specificity for bacteria over Hs would be a key consideration. Here it isn't. Or isn't explained well. In this context of cancer

treatment, specificity of action against other cellular/mitochondrial proteins, and *in vivo* effects of ClpP upregulation on tumour cells vs normal cells, would be far more useful criteria (for avoiding off-target effects).

Response: Thanks for the constructive comments. We reorganized the introduction to provide an explanation for why specificity for human over bacterium ClpP is desirable in this anticancer drug context.

As the amino acid sequences alignment showed (figure below for review only), *HsClpP* and ClpPs from the gut microbes (*Lactobacillus rhamnosus* GG, LGG; *Limosilactobacillus reuteri*, LR; *Bifidobacterium longum subsp. Infantis*, BL) and pathogenic bacteria (*S. aureus*, Sa and *E. coli*, Ec) are highly conserved in sequences. We assessed the chemo-activation effects of ClpP activators on the ClpPs from gut microbes and the inhibitory effects on the growth of the LGG, LR, and BL strains. The global activators for ClpP, ADEP 4, and ONC212 activated the recombinant *LrClpP in vitro* and inhibited the bacterium cell growth of LGG, LR, and BL strains (Fig. 2h and Fig. S2j-k). However, our selective *HsClpP* activator ZK53 did not activate ClpP from the microbe and minimally impaired the growth of LGG, LR, and BL strains (Fig. 2h and Fig. S2j-k). So, we postulate that selective *HsClpP* activators could help avoid the potential side effects on the gut microbes during the anticancer treatment. Supported by these data, we have added a sentence explaining the reason for specificity for human over bacterial ClpP in the introduction “Non-selective activators that should, for example, target *HsClpP* for anticancer effect could also activate ClpPs of gut microbes, which might cause negative effects during cancer therapy.”.

We agreed with the reviewer that in this context of cancer treatment, specificity of action against other cellular/mitochondrial proteins, and *in vivo* effects of ClpP upregulation on tumor cells vs. normal cells, would be far more useful criteria (for avoiding off-target effects). We assessed the inhibitory effects of our activators on the viability of non-malignant cells. ZK53 displayed less cytotoxicity on the human lung fibroblast MRC-5 cells than the LUSC tumor cells (Fig. S3f-g). While it should be

aware that the therapeutic window of ZK53 in LUSC treatment and the effects of ZK53 in non-malignant cells at therapeutic dosages still need more investigations in future.

Comments 4 from Reviewer 2: In the statement (Page 3 para 3) “In this study, we develop a highly selective and potent HsClpP activator ZK53”, it’s important to qualify selectivity as either protein or organism selectivity. Is selectivity only compared to *S. aureus* and *E. coli*, or to other organisms or proteins?

Response: We showed the selectivity of our activator ZK53 on *HsClpP* over the pathogenic bacterial ClpPs such as *SaClpP* and *EcClpP* (Fig. S1b) as well as the ClpP from the gut microbes such as *LrClpP* (Fig. S2j). In addition, ZK53 did not show inhibitory effects on cell growth in gut microbes, such as *Lactobacillus rhamnosus GG*; *Lactobacillus reuteri* and *Bifidobacterium longum infant* (Fig. 2h and Fig. S2k). Collectively, our study has shown the selectivity of ZK53 in both protein and organism among human, pathogenic bacteria, and gut microbes. With these data in hand, we

made a revision, “In this study, we develop a potent activator ZK53 that is highly selective on *HsClpP* but inactive toward bacterial ClpP proteins”.

Comments 5 from Reviewer 2: Could you define the proximity of the active site of the protease to the allosteric drug binding site in Figure 1 or 2 please.

Response: Thanks for the advice. We have shown the active site residues (S153, H178, and D227) of *HsClpP* in the right panel of Fig. 2a in red sticks and ZK53 in green sticks in the allosteric drug binding site. As the figure shows, the drug-binding site is far from the active site (approximately 26 Å).

Comments 6 from Reviewer 2: The authors have shown that ZK53 is much better at reducing tumour volume than Y118A. Given that Y118 is a peripheral part of the drug-binding pocket, it would seem relevant to understand whether it still binds the drug and if this is still effective. Inclusion of Y118A data in Fig 2f-g would strengthen the study.

Response: We thank the reviewer for the great suggestion. We have included the Y118A ClpP mutant data in Fig. S2g-i in the revision. We performed the NanoDSF assay to determine the thermal stability of Y118A ClpP mutant and ClpP WT in the presence of different activators. As shown in Fig. S2i, except ADEP 4, most activators displayed comparable thermally stabilized effects on ClpP WT and the Y118A ClpP mutant *in vitro*, which indicated ZK53 and the ONC201 analogs still bind to the Y118A ClpP mutant. In addition, we found that ZK53 is still effective to promote Y118A for α -casein degradation in PAGE assay (Fig. S2h), which is likely due to the constitutive activation of Y118A mutant itself (*Cancer Cell* **35**, 721-737 2019; *Cell Chem Biol*, **2022**, 29, 1396-408.e8). So, Y118 is not the key residue that is required for the binding and activation of ClpP by ZK53. We have added a statement in the revision, “Given that Y118 is a peripheral part of the ligand-binding pocket, it would seem relevant to understand whether the Y118A ClpP mutant still binds ZK53 and if this is still effective.”

Comments 7 from Reviewer 2: What do the authors mean (specifically) by ‘flexibility’

of ZK53? Do they mean free rotation about the single C-C bonds? In general, small molecules such as these have limited flexibility, adopting the lowest energy conformation, whether bound to a protein or not. It might be more accurate to say that ZK53 does not have a fused ring system (like imipridones), if that is what is meant.

Response: We appreciate the suggestion. We revised the statement to “ZK53 does not have a fused-ring system (like imipridones ONC201 and ONC212) or macrocyclic peptide scaffold (like ADEP 4)”.

Comments 8 from Reviewer 2: Following on, Page 14 states “we assume that ZK53 can freely approach the hydrophobic pockets of HsClpP and adopts a fixed configuration using an essential π - π stacking effect between the difluorobenzyl moiety and W146 of HsClpP”. If I am reading it correctly, this assumption is not consistent with knowledge of small molecules (as above).

Response: We have revised this sentence in revision, “When binding in the hydrophobic pockets, ZK53 adopts a fixed pose using π - π stacking effect between the difluorobenzyl moiety and W146 of *HsClpP*.”

Comments 9 from Reviewer 2: Page 3. “However, ADEPs also activated HsClpP and caused mammalian cell death”. This was stated as a defect, which it would be - if they were dealing with antibiotics. Discuss whether ZK53 also kill mammalian cells at therapeutic dosages? The EC₅₀ is close to that of ADEP 4 in Fig 2f. Have you tested ZF53 on non-xenografted control mice?

Response: It is right that the ADEPs antibiotics also activated *HsClpP* and caused mammalian cell death (*Cell Chem. Biol.* **25**, 1017-30 e9 2018), which indicates ADEPs are not safe antibiotics due to the nonselective activation among human and bacterial ClpPs. It is also true that the EC₅₀ of ZK53 is close to that of ADEP 4 in Fig 2f. So, we tested the antiproliferative effects of ZK53 on non-malignant cells. We found that ZK53 showed a three-fold higher IC₅₀ value in the human lung fibroblast MRC-5 cell line compared with the LUSC H1703 cell line (Fig. S3f). Additionally, ZK53 did not induce apoptosis in the MRC-5 cells (Fig. S3g). These data suggest relatively lower side effects

of ZK53 on the normal tissues.

In addition, we tested the biosafety of ZK53 (100 mg/kg, once a day, i.p) on non-xenografted control Balb/c mice. We did not observe body weight loss after treatment of ZK53 for seven days (please see the figure below for review only). Also, no significant body weight loss was observed after treatment of ZK53 for 28 days in the autochthonous mouse models (Fig. S6e).

With these data, we still could not exclude the potential side effects of ZK53 and others activators. So, we have added a discussion in the revision, “Although no obvious toxicity was observed in mouse models nor ZK53 inhibited gut microbes *in vitro*, our study does not present adequate information on therapeutic window of ZK53 nor the effects of ZK53 in more non-malignant cell lines at therapeutic dosages, which needs further investigations.”

Comments 10 from Reviewer 2: Page 6 and Fig. 2. The isoleucine mutant W146I does not bind ZK53 (by thermostability), due to an inability to pi-stack with the drug. This would seem to be of concern in terms of potential drug resistance. How conserved is that residue (W146) across eukaryotic species, and could this be a drawback of this simpler molecular scaffold with only a single critical interaction (π - π)? The Y118A mutant is evidence that the binding pocket can adapt.

Response: We agree with the reviewer that the hydrophobic binding pockets of human ClpP can adapt, which likely results in drug resistance. The residue W146 is highly conserved across eukaryotic species, including *Homo Sapiens*, *Mus Musculus*, and *Danio Rerio* (*Nat Commun*, **2022**, 13: 6909). It is possible that the critical role of W146 in ZK53 binding and activation causes potential drug resistance once W146 mutates. Again, this is a great point that should be addressed in future studies by monitoring the

rate of drug resistance induction. We have added a brief discussion in our revision in page 16, “It was reported that D190A mutation could be a mechanism of resistance to ClpP activator ONC201 (*Cancer Cell*, **2019**, 35, 721-37 e9). Probably, the critical role of W146 in ZK53 binding and activation might cause drug resistance once W146 mutates to less rigid amino acids.”.

Comments 11 from Reviewer 2: Page 9. The choice of proteins for Fig 4 appears limited to ETC members. If the choice is based on published data, a comparison with the literature is warranted. Otherwise, why are proteins from other mitochondrial compartments not also selected for testing? It would appear they have already decided what the answer may be (based on prior studies) and are not testing other proteins. Experimental choices made throughout are selective.

Response: Thanks for the great suggestion. Our choice of proteins was based on the previous works (*Cancer Cell*, **2019**, 35, 721-37 e9; *Cell Chem Biol*, **2022**, 29, 1396-408.e8). In the revision, we tested more mitochondrial proteins other than ETC proteins, including the mitochondrial transcription factor A (TFAM) and the mitochondrial protein elongation factor Tu (TUFM), which abundances were reduced upon treatment of ONC201, TR-57, and TR-107 (*Cancer Cell*, **2019**, 35, 721-37 e9; *ACS Chem Biol*, **2019**, 14, 1020-9; *Pharmacol Res Perspect*, **2022**, 10, e00993), and the TCA cycle enzyme aconitase (ACO2), which was reduced upon the treatment of TR-57 and TR-107 (*Pharmacol Res Perspect*, **2022**, 10, e00993). We also included the effects of ONC201 on these proteins. As shown in Fig. S4a, both treatments of our activator ZK53 and the anticancer drug candidate ONC201 similarly reduced the protein abundances of TFAM, TUFM, and ACO2 in H1703 cells in the immunoblot assay.

Comments 12 from Reviewer 2: Page 12. “The major organs in the endpoint of treatment were subjected to hematoxylin and eosin (H&E) staining, and no noticeable pathological changes were observed between the vehicle and ZK53-treated mice (Supplementary Fig. 6b).” Is this the case for the lung? The vehicle and ZK53 histology are significantly different.

Response: We took the lung H&E images from different locations, which led to this discrepancy. We have replaced the lung image of the vehicle group in Fig. S6c. We have also scanned the whole lung tissue, as shown in the figure for review only below, and there is no significant difference between the vehicle and ZK53 group.

Comments 13 from Reviewer 2: The heading of Supplementary. Fig 6 ‘ZK53 exhibits antitumor activity in vivo’ doesn't reflect the data shown. The caption requires much more detail and should state if these are all xenograft mice.

Response: We are sorry for writing this statement inaccurately. We added new data in Fig. S6, and now we changed the heading of Fig. S6 to “ZK53 shows anticancer effect and low toxicity in xenograft and the genetically engineered mouse models.”.

Comments 14 from Reviewer 2: The PDB report supplied is a preliminary one – annotated “Not for Manuscripts”, indicating a full deposition has not yet been made. Importantly, the ligands (ZK53) central to the manuscript are not included. A final deposition report should be supplied.

Response: We have completed the deposition of ZK53/ClpP structure in PDB, and now the formal final validation report has been provided in our revised submission. The ligand ZK53 is referred as ‘ZLL’ in the report.

Comments 15 from Reviewer 2: Given the high apparent symmetry of the complex (two heptamers), was there a reason for individual refinement of the 14 chains throughout (without applying NCS averaging at any stage)?

Response: Thanks for the question. The structure was refined using the re mac5 program of the CCP4i suite, and automatic local NCS restraint was applied during the refinement. Please see below for some of the local NCS restraints.

REMARK	3	NCS RESTRAINTS STATISTICS									
REMARK	3	NCS TYPE: LOCAL									
REMARK	3	NUMBER OF DIFFERENT NCS PAIRS: 91									
REMARK	3	GROUP	CHAIN1	RANGE	CHAIN2	RANGE	COUNT	RMS	WEIGHT		
REMARK	3	1	A	58 249	B	58 249	10954	0.02	0.05		
REMARK	3	2	A	58 249	C	58 249	10918	0.06	0.05		
REMARK	3	3	A	58 249	D	58 249	10920	0.06	0.05		
REMARK	3	4	A	58 249	E	58 249	11164	0.05	0.05		
REMARK	3	5	A	58 249	F	58 249	10922	0.07	0.05		
REMARK	3	6	A	58 249	G	58 249	10972	0.05	0.05		
REMARK	3	7	A	58 248	H	58 248	10732	0.05	0.05		
REMARK	3	8	A	58 249	I	58 249	10896	0.07	0.05		
REMARK	3	9	A	57 248	J	57 248	10856	0.05	0.05		
REMARK	3	10	A	58 249	K	58 249	10928	0.06	0.05		
REMARK	3	11	A	58 249	L	58 249	10962	0.05	0.05		
REMARK	3	12	A	58 248	M	58 248	11026	0.05	0.05		
REMARK	3	13	A	58 248	N	58 248	10954	0.06	0.05		

Comments 16 from Reviewer 2: The methods section should detail the protocols used in refinement and specify particulars of the model - such as chain breaks.

Response: Thanks for the suggestion. Detailed information has been provided in the method section. “The structure was refined using the *refmac5* program of the CCP4i suite. During the refinement, residues with no electron density were removed, and local NCS restraints were automatically applied to the fourteen protomers within the model.”

Comments 17 from Reviewer 2: For a structure refined to 1.9 Å resolution, the B-factors are a bit higher than would be expected, and I would also have expected more water molecules built in. Could you include examples of the electron density maps in the Supplementary Information please.

Response: Thanks for reminding us of the relatively high B-factor of the structure. *HsClpP* exists as a dimer of two heptamers. As depicted in panel A in the figure below, the *HsClpP* structure has a big channel in the center. The packing of the *HsClpP* structure is also very loose (please see panel B in the figure below for review only). Residues located in the central channel and the interfaces all have a higher B-factor,

which may lead to the relatively high B-factor of the structure.

We agree with the reviewer that the number of water molecules is relatively small. We have carefully checked the electron density maps of our structure. Unfortunately, no obvious electron density for water molecules could be found. We have included the electron density map of the ligand-binding area of chain A in Fig. S2a.

Comments 18 from Reviewer 2: The report indicates a 2.4% twinning fraction. Was this treated appropriately?

Response: Our structure was refined using the Refmac5 program. During the initial refinement, we tested the possibility of twinning of the data. Consistent with the validation report, no obvious twinning fraction was detected by the Refmac5 program. Therefore, we did not impose twinning during the final refinement of the structure.

3. Response to the comments of Reviewer 3:

General Comments from Reviewer 1: In this work, the authors report the development of ZK53, a specific activator of HsClpP, where the crystal structure ZK53/ClpP showed the importance of W146 residue of ClpP in the selective binding with the difluorobenzyl motif of the activator. The mode of action of ZK53 was further assessed by showing the inhibition of LUSC cell proliferation, by targeting the ETC subunits of

mitochondrial proteins. Additional relevant data supported the effect of ZK53 on the essential regulatory pathways for G1/S transition, and its activation of the ATM-mediated DNA damage. Finally ZK53 was shown to exhibit anticancer activity in in vivo mice model. Collectively, all the data reported provided enough evidence regarding the druggability of HsClpP as an anticancer target in LUSC therapy. I recommend this paper to be accepted.

Response: We thank the reviewer for recommendation for publication.

Comments from Reviewer 3: I do only suggest in the discussion part, to provide clearer points of comparison between ZK53 and other ClpP activators, such as ONC201, ADEP 28, for example in a table form, with the precise reported mechanism of action, selectivity, whether flexible or rigid scaffold,.. just to emphasize the similarities and differences between ZK53 and the other activators

Response: Thank the reviewer for the great suggestion. We have added a table (Table S6) describing the comparisons between ZK53 and other ClpP activators in discussion. We have added several sentences to clarify this point, “ZK53 does not have a fused-ring system (like imipridones ONC201 and ONC212) or macrocyclic peptide scaffold (like ADEP 4), which is distinct from those *HsClpP* activators typically sharing rigid scaffolds to achieve strong binding affinity (Table S6). The similarities and differences between ZK53 and other activators will facilitate further exploration of the druggability of *HsClpP* as an anticancer target.”

4. Response to the comments of Reviewer 4:

General Comments from Reviewer 1: The study by Zhou et. al. entitled by “Selective activator of human ClpP triggers cell cycle arrest to inhibit lung squamous cell carcinoma” describes a newly developed activator of caseinolytic proteinase P (ClpP), ZK53 that exhibits surprisingly strong anticancer effects in a human lung squamous xenograft model. Given its well-established role for mitochondrial proteome homeostasis, ZK53 appears to promote the degradation of components of mitochondrial electron transfer chain resulting in disruption of oxidative respiration

and ATP generation in squamous cancer cells. The authors further demonstrate that ZK53 suppresses E2F target network and induces cell-cycle arrest although the molecular mechanism underlying the connection between ZK53-mediated mitochondrial alteration and E2F signaling is unclear.

Response: Thank the reviewer for the positive comments and constructive suggestions. We have carefully revised our manuscript and provided a point-by-point response to the specific comments below.

Comments 1 from Reviewer 4: Considering therapeutic potential of anti-OXPHOS for cancer therapeutics, this study provides a clinically relevant proof-of-concept of targeting mitochondrial bioenergetic functions for cancer therapy. However, a recent study that reported serious neurological adverse effects in patients treated with complex I inhibitor (IACS-010759, Yap et. al. Nat Med, 29, 115-126, 2023) significantly weakens the enthusiasm of this study. Although the authors argue that no obvious toxicities have been noted in histology and blood biochemical analysis, this study does not present information on therapeutic window of ZK53 nor the effects of ZK53 in non-malignant cells (e.g. human bronchial epithelial cells). It would be critical to show ZK53 has significantly less or no mitochondrial OXPHOS inhibition as well as cytotoxic effects on normal tissues.

Response: We agree with the reviewer. We have assessed the cytotoxic effects of ZK53 on the human lung fibroblast MRC-5 cell line and found that ZK53 showed a three-fold higher IC₅₀ value of MRC-5 compared with the LUSC cell H1703 (Fig. S3f). Additionally, ZK53 did not significantly induce cell apoptosis in MRC-5 cells (Fig. S3g). ZK53 also shows less OXPHOS inhibition as evidenced by no significant decrease of the ATP level in MRC-5 cells (Fig. S4e). These results suggest relatively weak effects of the ClpP activator ZK53 on normal tissues. We agreed with the reviewer that with these data we still could not exclude the side effects of ZK53 and more investigations on biosafety and the therapeutic window of ZK53 for LUSC therapy should be carefully explored in future studies. So, we added a discussion in the revision, “Although no obvious toxicity was observed in mouse models nor ZK53 inhibited gut

microbes *in vitro*, our study does not present adequate information on therapeutic window of ZK53 nor the effects of ZK53 in more non-malignant cell lines at therapeutic dosages, which needs further investigations.” We also cited this literature (Yap et. al. *Nature Medicine*, **29**, 115-126, 2023) in the first paragraph of discussion.

Comments 2 from Reviewer 4: Another critical weakness of his study is to depend entirely on cell line system, which significantly weakens the clinical relevance. It would be extremely interest to validate its anticancer effects on autochthonous mouse models such as *Kras*^{G12D}; *Lkb1*-null or *Lkb1*-null; *PTEN*-null mice. These two systems have been well-employed by a number of studies and reported to develop lung squamous carcinoma with other types of lung cancers.

Response: We thank the reviewer for the great suggestion. We have evaluated the anticancer effect of our activator ZK53 on the *Kras*^{LSL-G12D/+}; *Lkb1*^{fl/fl} (KL) genetically engineered mouse model (GEMM). The data in Fig. 6f-h showed that ZK53 treatment significantly suppressed LUSC progression in the autochthonous mouse model. The Ki67 immunostaining results further showed that LUSC cell proliferation was noticeably suppressed in the ZK53-treated group (Fig. 6i-j). Moreover, we observed no significant loss of body weight in the ZK53-treated group compared with vehicle control (Fig. S6e).

Comments 3 from Reviewer 4: It is unclear why lung squamous cancer has been selected for this study. Is lung squamous cancer most sensitive to ZK53? If so, what would be underlying mechanisms for this metabolic vulnerability? The authors need to provide biological as well as clinical rationale for lung squamous cancers as the best candidate for ZK53.

Response: Thanks for the great suggestion. We have reorganized our introduction to provide an explanation why LUSC has been selected for this study. Because LUSC accounts for 25-30% of lung carcinoma death, LUSC is less benefited from targeted therapy such as EGFR inhibitor, which is different from lung adenocarcinoma. Additionally, the response to immunotherapy is also low. So, there is an unmet clinical

need for LUSC treatment. Using the analysis of TCGA database (Fig. 3d) and the Kaplan-Meier plotter (please see the figure below), we observe elevated expression levels in LUSC tumor tissues and prolonged overall survival in patients with higher ClpP levels, which indicates the importance of ClpP in LUSC development. In addition, ClpP activators have been shown to be effective in multiple cancer treatments, especially the imipridone ONC201 and the derivatives have been entered into late phase clinal trial. However, there are no reports on the therapeutic potential of ClpP activator on LUSC cancer. So, these promoted us to select LUSC as our candidate cancer type in this study. We agree with the reviewer that it is important to evaluate the sensitivity of lung squamous cancer to our activator compared with other cancers in the future.

We are aware of the limitations in this work and we tried to discuss in revision, “In addition, it remains unclear about the tumor biology as well as underlying mechanisms for this metabolic vulnerability for LUSC cancers as the best candidate for certain ClpP activator, such as ZK53.” In page 17.

REVIEWER COMMENTS

Reviewer #1 (Remarks to the Author):

I have thoroughly reviewed the author's response and the revised manuscript. The author has diligently addressed the raised issues and incorporated suggested improvements, which significantly enhance the quality and value of their research. After careful consideration, I concur that the paper is now suitable for publication in the prestigious journal Nature Communications.

Reviewer #2 (Remarks to the Author):

Overall, the manuscript improved upon resubmission, and we appreciate the effort the authors have gone to with provision of additional data for comparison, points of clarification, removal of overstatements, and inclusion of preliminary biosafety data. The responses throughout were largely satisfactory.

The rationale provided for selectivity of human ClpP vs gut microbe ClpP is reasonable. Importantly, the authors assessed effects of the drugs on viability of non-malignant cells, finding lower toxicity in these than in the tumour line.

General Comments:

Comment 1. Suppl. Fig. 2h – please indicate duration time used in Figure caption, for comparison to part g. Also include this for Suppl 2c, d, f.

Comment 8:

“When binding in the hydrophobic pockets, ZK53 adopts a fixed pose using n-n stacking effect between the difluorobenzyl moiety and W146 of HsClpP.”

Suggested change to:

“In the hydrophobic pocket of HsClpP, the 3,5-difluorophenyl moiety of ZK53 makes n-stacking interactions with the indole ring of W146.”

Comment 10:

“Probably, the critical role of W146 in ZK53 binding and activation might cause drug resistance once W146 mutates to less rigid amino acids.”.

Suggested change to: “The dependency of ZK53 binding and activation on a single residue (W146) may be a factor in development of drug resistance”.

The questions on structure were not addressed satisfactorily. Please correct as below:

The electron density shown in the rejoinder is not informative. The views provided are not close-ups. The density at a glance appears less featured than expected for $\sim 1.9 \text{ \AA}$, which would explain the inability to distinguish water molecules. Also, Suppl Fig 2a is pixelated; please rectify. To demonstrate map quality, please include some close-ups of density at the core and the periphery of a monomer in figures (Supps?).

Figure 2: Binding pocket.

I suggest replacing Fig. 2c with a stereo-view, with ALL relevant residues (including those in fig 2d) to show the base stacking and overall fit of the ZK53.

I also suggest a panel showing the electron density for ZK53 in situ in the binding site, along with that of key residues involved in binding (e.g. Q, W, Y etc).

In Fig 2d, the interaction with W146 appears to be the middle ring of ZK53. This does not agree with the structure. Please check the interactions are as sketched.

Also, what are the 'eyelashes' on the aromatic rings of ZK53? Please define.

Why is E82 depicted as hydrophobic in Fig 2d?

Is it the alkyl chain of Glu that stacks? It would be helpful to see the residue in Fig. 2c.

Final PDB report:

Curiously, some of the residues depicted in Figs 2c-d (e.g. E82, L148, L170) correspond to the strongest of the rotamer outliers in the PDB report provided. L170 is flagged as non-rotameric. This is suggestive (but by no means proof) of deformation of the binding pocket when ZK53 binds. Did you compare the binding pocket with that of your starting model, the apo-HsClpP structure (1TG6)? Any significant deformation should be mentioned in the manuscript (binding site changes can factor into drug resistance).

Other points:

The diffraction data quality is excellent, based on Suppl. Table 1; the density maps appear less so. I would be curious to see the final data refinement statistics in resolution shells, from the data processing output of HKL2000, and a molecular packing diagram (unit cell) with axes labelled.

Where the text was amended, rather than saying "residues with no electron density were removed" it is usual to provide some information re the structure, e.g. 'residues x to y of each chain were modelled', or 'xx residues at the C-terminus/N-terminus/or xx to yy in specific loops were omitted from the model due to poorly ordered electron density'. The following residues were truncated at C β (or include a table in the supplementary data).

Further, it is usual to state in the Methods whether the entire tetradecamer of apo-HsClpP was used as a search model, and whether there is an entire complex in the asymmetric unit of your structure.

Reviewer #4 (Remarks to the Author):

The KrasG12D; LKB1^{-/-} (KL) GEMM model produces a mixture of lung adenocarcinoma, squamous cell carcinoma, and large cell carcinoma. According to the literature, the major tumor type developed in KL GEMM mice is lung adenocarcinoma. Squamous cell carcinoma accounts for approximately 25-30% of all tumors in the lungs of KL mice. This classification can be determined through pathological evaluation and further confirmed by conducting immunohistochemical analysis using squamous cancer markers. These markers include p63 (positive) and TTF-1 (negative), which are used to differentiate squamous cell carcinoma from lung adenocarcinoma.

However, the newly added experimental results using KL mice lack this crucial information. Therefore, it is imperative to demonstrate that the ZK53 treatment specifically reduces the growth of squamous cell carcinoma while leaving the progression of lung adenocarcinoma unaffected. To validate the specific effects of ZK53 on squamous cell carcinoma, it is recommended to present immunohistochemical analysis results.

To support the findings of the experiment, it is necessary to provide evidence showing the selective impact of ZK53 on squamous cell carcinoma. This can be achieved through immunohistochemical analysis, which will validate the distinct effects of ZK53 on squamous cell carcinoma while not affecting the progression of lung adenocarcinoma.

Response to the reviewers' comments

Selective activator of human ClpP triggers cell cycle arrest to inhibit lung squamous cell carcinoma

NCOMMS-22-48226A

Reviewer #1:

General comments from Reviewer 1: I have thoroughly reviewed the author's response and the revised manuscript. The author has diligently addressed the raised issues and incorporated suggested improvements, which significantly enhance the quality and value of their research. After careful consideration, I concur that the paper is now suitable for publication in the prestigious journal Nature Communications.

Response: We thank the reviewer for recommendation.

Reviewer #2:

General comments from Reviewer 1: Overall, the manuscript improved upon resubmission, and we appreciate the effort the authors have gone to with provision of additional data for comparison, points of clarification, removal of overstatements, and inclusion of preliminary biosafety data. The responses throughout were largely satisfactory. The rationale provided for selectivity of human ClpP vs gut microbe ClpP is reasonable. Importantly, the authors assessed effects of the drugs on viability of non-malignant cells, finding lower toxicity in these than in the tumour line.

Response: We thank the reviewer for the constructive and positive comments.

Comment 1 from Reviewer 2: Suppl. Fig. 2h – please indicate duration time used in Figure caption, for comparison to part g. Also include this for Suppl 2c, d, f.

Response: The reaction time, 2 h, was included in the legends of these Suppl. figures.

Comment 2 from Reviewer 2: “When binding in the hydrophobic pockets, ZK53 adopts a fixed pose using π - π stacking effect between the difluorobenzyl moiety and W146 of HsClpP.”

Suggested change to: “In the hydrophobic pocket of HsClpP, the 3,5-difluorophenyl moiety of ZK53 makes π -stacking interactions with the indole ring of W146.”

Response: We changed the previous sentence to the recommended one.

Comment 3 from Reviewer2: “Probably, the critical role of W146 in ZK53 binding and activation might cause drug resistance once W146 mutates to less rigid amino acids.”.

Suggested change to: “The dependency of ZK53 binding and activation on a single residue (W146) may be a factor in development of drug resistance”.

Response: We changed this sentence with the recommended one.

Comment 4 from Reviewer2: The questions on structure were not addressed satisfactorily. Please correct as below. The electron density shown in the rejoinder is not informative. The views provided are not close-ups. The density at a glance appears less featured than expected for ~ 1.9 Å, which would explain the inability to distinguish water molecules. Also, Suppl Fig 2a is pixelated; please rectify. To demonstrate map quality, please include some close-ups of density at the core and the periphery of a monomer in figures (Supps?).

Response: Following the reviewer’s suggestion, new images showing the electron density of $\alpha 2$ and $\alpha 4$ of *HsClpP* were included in Suppl. Fig. 2b in the revised manuscript. $\alpha 2$ and $\alpha 4$ locate at the periphery and core of *HsClpP*, respectively (Suppl. Fig. 2a). The previous Suppl. Fig. 2a (now Suppl. Fig. 2c) in high resolution was provided in the revised manuscript.

Comment 5 from Reviewer2: Figure 2: Binding pocket. I suggest replacing Fig. 2c with a stereo-view, with ALL relevant residues (including those in fig 2d) to show the base stacking and overall fit of the ZK53.

Response: We agree with the reviewer. Fig. 2c in a stereo-view, with ALL relevant residues (including those in previous Fig. 2d) was provided to show the base stacking and overall fit of the ZK53. And the previous Fig. 2d showing the hydrophobic interactions between ZK53 and the *HsClpP* was omitted in the revision.

Comment 6 from Reviewer2: I also suggest a panel showing the electron density for ZK53 in situ in the binding site, along with that of key residues involved in binding (e.g. Q, W, Y etc).

Response: We replaced the previous Fig. 2b with a new image to show the electron density for ZK53 in situ in the binding site, along with that of key residues involved in binding.

Comment 7 from Reviewer2: In Fig 2d, the interaction with W146 appears to be the middle ring of ZK53. This does not agree with the structure. Please check the interactions are as sketched.

Response: We agree with the reviewer and omitted the previous Fig. 2d in the revision, because the revised Fig. 2c includes ALL relevant residues (including those in previous Fig. 2d).

Comment 8 from Reviewer2: Also, what are the 'eyelashes' on the aromatic rings of ZK53? Please define.

Response: We provide a new Fig. 2c to include those in previous Fig. 2d.

Comment 9 from Reviewer2: Why is E82 depicted as hydrophobic in Fig 2d? Is it the alkyl chain of Glu that stacks? It would be helpful to see the residue in Fig. 2c.

Response: We have presented a new Fig. 2c in a stereo-view that shows ZK53 binding in the hydrophobic site, along with that of key residues involved in binding. The alkyl chain of E82 stacks with ZK53, which feature is also included in Fig. 2c.

Comment 10 from Reviewer2: Final PDB report: Curiously, some of the residues depicted in Figs 2c-d (e.g. E82, V148, L170) correspond to the strongest of the rotamer outliers in the PDB report provided. L170 is flagged as non-rotameric. This is suggestive (but by no means proof) of deformation of the binding pocket when ZK53 binds. Did you compare the binding pocket with that of your starting model, the apo-

HsClpP structure (1TG6)? Any significant deformation should be mentioned in the manuscript (binding site changes can factor into drug resistance).

Response: We did notice that some of the residues (E82, V148, L170) correspond to the strongest of the rotamer outliers in the PDB validation report, and L170 is flagged as non-rotameric. However, these residues show good electron density maps as depicted in Fig 2b. Also, following the reviewer's suggestion, we compared the apo-HsClpP and ZK53-bound structures. As depicted in Suppl. Fig. 2e in the revised manuscript, no large conformational change was observed for E82, whereas ZK53 binding leads to rotation of the side chains of L79, V148 and L170. We have added a statement in the revision.

Comment 11 from Reviewer2: Other points: The diffraction data quality is excellent, based on Suppl. Table 1; the density maps appear less so. I would be curious to see the final data refinement statistics in resolution shells, from the data processing output of HKL2000, and a molecular packing diagram (unit cell) with axes labelled.

Response: We are sorry that in the previous submission we referred to the wrong data processing program. Instead of HKL2000, we processed the diffraction data of the ZK53/HsClpP complex using Aquarium, an automatic data-processing and experiment information management system for biological macromolecular crystallography beamlines (*J. Appl. Cryst.*, **2019**, 52, 472-477). We uploaded the final log file (LDX004-12_1_xia2_3dii.log) for this review only. We also included two representative data screenshots in this response for convenience.

\$TABLE: Analysis against resolution, with & without anomalous (Ov), SAD:
\$GRAPHS:Rmerge, Rmeas, Rpm v Resolution:0|0.277008x0|2.006:2,4,5,8,9,10,11:

\$\$												
N	1/d ²	Dmid	Rmrg	RmrgOv	Rcum	RcumOv	Rmeas	RmeasOv	Rpm	RpmOv	Nmeas	\$\$ \$\$
1	0.0069	12.02	0.043	0.047	0.043	0.047	0.051	0.051	0.027	0.019	15305	
2	0.0208	6.94	0.050	0.054	0.046	0.050	0.058	0.059	0.030	0.022	32023	
3	0.0346	5.37	0.052	0.058	0.048	0.052	0.062	0.063	0.033	0.024	39081	
4	0.0485	4.54	0.049	0.054	0.048	0.053	0.059	0.059	0.032	0.023	45875	
5	0.0623	4.01	0.054	0.062	0.050	0.055	0.066	0.068	0.036	0.027	48917	
6	0.0762	3.62	0.064	0.071	0.052	0.058	0.076	0.077	0.041	0.030	58212	
7	0.0900	3.33	0.083	0.092	0.055	0.061	0.099	0.100	0.053	0.038	65599	
8	0.1039	3.10	0.107	0.117	0.058	0.064	0.126	0.126	0.067	0.048	71846	
9	0.1177	2.91	0.148	0.161	0.062	0.068	0.175	0.173	0.092	0.065	77657	
10	0.1316	2.76	0.199	0.214	0.066	0.073	0.234	0.231	0.123	0.087	82265	
11	0.1454	2.62	0.249	0.268	0.070	0.077	0.293	0.289	0.154	0.108	86623	
12	0.1593	2.51	0.323	0.355	0.074	0.082	0.386	0.387	0.209	0.151	81843	
13	0.1731	2.40	0.426	0.458	0.079	0.087	0.506	0.496	0.270	0.189	90862	
14	0.1870	2.31	0.524	0.565	0.084	0.092	0.628	0.616	0.342	0.241	87886	
15	0.2008	2.23	0.619	0.665	0.088	0.097	0.738	0.722	0.399	0.279	94610	
16	0.2147	2.16	0.755	0.804	0.093	0.102	0.897	0.870	0.479	0.331	101340	
17	0.2285	2.09	0.855	0.906	0.098	0.107	1.011	0.979	0.536	0.369	105686	
18	0.2424	2.03	1.026	1.083	0.102	0.112	1.211	1.170	0.638	0.438	110538	
19	0.2562	1.98	1.350	1.427	0.106	0.116	1.591	1.540	0.836	0.576	114035	
20	0.2701	1.92	1.702	1.810	0.110	0.120	2.006	1.953	1.053	0.728	117526	
\$\$												
Overall:			0.110	0.120	0.110	0.120	0.131	0.131	0.070	0.050	1527729	
N	1/d ²	Dmid	Rmrg	RmrgOv	Rcum	RcumOv	Rmeas	RmeasOv	Rpm	RpmOv	Nmeas	

\$TABLE: Completeness & multiplicity v. resolution, SAD:
\$GRAPHS:Completeness v Resolution :N:2,7,8,10,11:
:Multiplicity v Resolution:0|0.277008x0|7.15037:2,9,12:

\$\$												
N	1/d ²	Dmid	Nmeas	Nref	Ncent	%poss	C%poss	Mlplct	AnoCmp	AnoFrc	AnoMlt	\$\$ \$\$
1	0.0069	12.02	15306	2398	273	91.9	91.9	6.4	89.4	96.7	3.5	
2	0.0208	6.94	32023	4634	323	99.5	96.8	6.9	98.2	98.6	3.6	
3	0.0346	5.37	39084	5969	317	99.6	98.1	6.5	97.5	97.8	3.4	
4	0.0485	4.54	45875	7043	318	99.8	98.7	6.5	98.3	98.5	3.3	
5	0.0623	4.01	48927	7984	303	99.7	99.0	6.1	98.2	98.3	3.1	
6	0.0762	3.62	58215	8772	319	99.8	99.2	6.6	98.5	98.6	3.4	
7	0.0900	3.33	65601	9569	320	100.0	99.4	6.9	99.1	99.2	3.5	
8	0.1039	3.10	71847	10282	321	100.0	99.5	7.0	99.3	99.3	3.6	
9	0.1177	2.91	77658	10939	319	100.0	99.6	7.1	99.5	99.5	3.6	
10	0.1316	2.76	82265	11505	321	100.0	99.6	7.2	99.5	99.5	3.6	
11	0.1454	2.62	86623	12116	330	100.0	99.7	7.1	99.5	99.5	3.6	
12	0.1593	2.51	81844	12667	322	100.0	99.7	6.5	99.5	99.5	3.3	
13	0.1731	2.40	90865	13235	323	100.0	99.7	6.9	99.5	99.5	3.5	
14	0.1870	2.31	87890	13742	323	100.0	99.8	6.4	99.7	99.7	3.2	
15	0.2008	2.23	94624	14214	323	100.0	99.8	6.7	99.7	99.7	3.4	
16	0.2147	2.16	101340	14698	327	100.0	99.8	6.9	99.8	99.8	3.5	
17	0.2285	2.09	105687	15115	324	100.0	99.8	7.0	99.8	99.8	3.5	
18	0.2424	2.03	110540	15625	320	100.0	99.8	7.1	99.9	99.9	3.6	
19	0.2562	1.98	114035	16043	330	100.0	99.9	7.1	99.9	99.9	3.6	
20	0.2701	1.92	117526	16466	322	100.0	99.9	7.1	99.9	99.9	3.6	
\$\$												
Overall:			1527775	223016	6378	99.9	99.9	6.9	99.3	99.4	3.5	
			Nmeas	Nref	Ncent	%poss	C%poss	Mlplct	AnoCmp	AnoFrc	AnoMlt	

The packing of *HsClpP*-ZG53 complex in the crystal lattice is shown below.

Comment 12 from Reviewer2: Where the text was amended, rather than saying “residues with no electron density were removed” it is usual to provide some information re the structure, e.g. ‘residues x to y of each chain were modelled’, or ‘xx residues at the C-terminus/N-terminus/or xx to yy in specific loops were omitted from the model due to poorly ordered electron density’. The following residues were truncated at C β (or include a table in the supplementary data).

Response: Thanks for the great suggestion. We have included the structural information and added a sentence in page 5, “The residues at the terminus or in specific loops, and some side chains were omitted from the model due to poorly ordered electron density (Supplementary Table 2).”.

Comment 13 from Reviewer2: Further, it is usual to state in the Methods whether the entire tetradecamer of apo-HsClpP was used as a search model, and whether there is an entire complex in the asymmetric unit of your structure.

Response: Thanks for the suggestion. The reported apo-HsClpP structure (PDB_ID: 1TG6) and our HsClpP-ZK53 complex belong to different space group. In our structural complex, per asymmetric unit contains one entire tetradecamer of HsClpP. In the reported apo-HsClpP structure, per asymmetric unit contains one heptamer of HsClpP, which was directly utilized as the search model in our structural determination. To

clarify this concern, we have revised the experimental method to “The structure was solved by the molecular replacement method using the Phaser program embedded in the CCP4i suite with the entire heptamer in *HsClpP* structure (PDB: 1TG6) as the search model” in page 21.

Reviewer #4:

General comments from Reviewer 4: The *Kras*G12D; *LKB1*^{-/-} (KL) GEMM model produces a mixture of lung adenocarcinoma, squamous cell carcinoma, and large cell carcinoma. According to the literature, the major tumor type developed in KL GEMM mice is lung adenocarcinoma. Squamous cell carcinoma accounts for approximately 25-30% of all tumors in the lungs of KL mice. This classification can be determined through pathological evaluation and further confirmed by conducting immunohistochemical analysis using squamous cancer markers. These markers include p63 (positive) and TTF-1 (negative), which are used to differentiate squamous cell carcinoma from lung adenocarcinoma.

However, the newly added experimental results using KL mice lack this crucial information. Therefore, it is imperative to demonstrate that the ZK53 treatment specifically reduces the growth of squamous cell carcinoma while leaving the progression of lung adenocarcinoma unaffected. To validate the specific effects of ZK53 on squamous cell carcinoma, it is recommended to present immunohistochemical analysis results.

To support the findings of the experiment, it is necessary to provide evidence showing the selective impact of ZK53 on squamous cell carcinoma. This can be achieved through immunohistochemical analysis, which will validate the distinct effects of ZK53 on squamous cell carcinoma while not affecting the progression of lung adenocarcinoma.

Response: We thank the reviewer for the great suggestion. Following the reviewer’s suggestion, we analyzed the progression of mouse LUAD, as distinguished by histopathological analyses and immunostaining analyses of LUAD marker TTF1 and LUSC marker p40 (Suppl. Fig. 6e). Statistical analyses showed no significant changes in

LUAD tumor burden and number with ZK53 treatment (Suppl. Fig. 6f, g), suggesting that ZK53 indeed inhibited LUSC rather than LUAD. We have supplemented these results in our revision.

REVIEWER COMMENTS

Reviewer #2 (Remarks to the Author):

The authors have put in significant effort and addressed all of the points raised to my satisfaction.

Reviewer #4 (Remarks to the Author):

The reviewer believes that the authors have not successfully addressed issues raised by the reviewer, including the issues regarding the analysis of KL murine model (which should be very straightforward). The reviewer suggests that the authors refer to the following articles.

<https://www.sciencedirect.com/science/article/pii/S2211124719309246?via=ihub>

<https://www.nature.com/articles/s41586-023-05793-3>

Response to the reviewers' comments

Selective activator of human ClpP triggers cell cycle arrest to inhibit lung squamous cell carcinoma

NCOMMS-22-48226B

Reviewer #4:

The reviewer believes that the authors have not successfully addressed issues raised by the reviewer, including the issues regarding the analysis of KL murine model (which should be very straightforward). The reviewer suggests that the authors refer to the following articles.

<https://www.sciencedirect.com/science/article/pii/S2211124719309246?via=ihub>

<https://www.nature.com/articles/s41586-023-05793-3>

Response: Thanks for this great suggestion. In our previous revision, we presented the effect of ZK53 on the tumor burden of LUAD and LUSC separately, one in the Supplementary figure and the other in the main figure. As suggested, we have now merged these two analyses into one figure, as shown in the manuscript as well as the Fig. 6f-g attached here. Also, we have cited these two references in revised manuscript. To make it easy to observe the specific effect of ZK53 treatments upon LUSC but not LUAD progression, we chose to show the percentage of LUAD/LUSC burdens in whole lungs (new Fig. 6f-g) instead of showing the ratio of LUAD/LUSC in total lung tumors (Referee only Fig. 1). As you can see from the Referee only Fig. 1, it could lead to some confusion if we present the ratio of LUAD/LUSC.

New Fig. 6. f Representative H&E staining and immunohistochemistry staining for TTF1 and p40 in LUAD and LUSC from KL mice treated with vehicle or ZK53. LUAD is indicated by blue dashed

line, and LUSC is presented by red dashed line. Scale bar for whole lung: 1 mm; Scale bar for others: 50 μ m. **g** Quantification of ratios of LUAD, LUSC, and normal tissue in the whole lungs of KL mice treated with vehicle or ZK53.

Referee only Fig. 1 Quantification of LUAD and LUSC tumor burden in KL mice treated with vehicle or ZK53.

We also want to mention that, different from previous report showing the ratio of LUSC/LUAD at 25-30%, our mouse cohort displayed a relatively higher LUSC/LUAD ratio at $40.58 \pm 13.46\%$ (Referee only Fig. 1). The discrepancy might be attributed to different experimental settings, e.g., mouse genetic background, virus delivery methods...etc.

REVIEWERS' COMMENTS

Reviewer #4 (Remarks to the Author):

Issues the reviewer has raised have been well addressed.